# Bayesian integration of flux tower data into process-based simulator for quantifying uncertainty in simulated output

Rahul Raj[1], Christiaan van der Tol[1], Alfred Stein[1], and Nicholas A.S. Hamm[1]

[1]Faculty of Geo-Information Science and Earth Observation (ITC), University of Twente, PO Box 217, 7514 AE Enschede, The Netherlands

*Correspondence to:* Rahul Raj (r.raj@utwente.nl, rahulosho@gmail.com)

**Abstract.**

Parameters of a process-based forest growth simulator are difficult or impossible to obtain from field observations. Reliable estimates can be obtained using calibration against observations of output and state variables. In this study, we present a Bayesian framework to calibrate the widely used process-based simulator BIOME-BGC against estimates of gross primary
production (GPP) data. We used GPP partitioned from flux tower measurements of a net ecosystem exchange over a 55 year old Douglas fir stand as an example. The uncertainties of both the BIOME-BGC parameters and the simulated GPP were estimated. The calibrated parameters leaf and fine root turnover (LFRT), ratio of fine root carbon to leaf carbon (FRC:LC), ratio of carbon to nitrogen in leaf (C:N$_{leaf}$), canopy water interception coefficient ($W_{int}$), fraction of leaf nitrogen in Rubisco (FLNR), and soil rooting depth (SD) characterize the photosynthesis and carbon and nitrogen allocation in the forest. The
calibration improved the root mean square error and enhanced Nash-Sutcliffe efficiency between simulated and flux tower daily GPP compared to the uncalibrated BIOME-BGC. Nevertheless, the seasonal cycle for flux tower GPP was not reproduced exactly, and some overestimation in spring and underestimation in summer remained after calibration. We hypothesized that the phenology exhibited a seasonal cycle that was not accurately reproduced by the simulator. We investigated this by calibrating the BIOME-BGC to each month's flux tower GPP separately. As expected, the simulated GPP improved, but the calibrated
parameter values suggested that the seasonal cycle of state variables in the simulator could be improved. It was concluded that the Bayesian framework for calibration can reveal features of the modelled physical processes, and identify aspects of the process simulator that are too rigid.

Keywords: Process-based simulator, BIOME-BGC, gross primary production, Bayesian calibration, uncertainty estimation.

## 1 Introduction

Forest ecosystems play an important role in the global carbon cycle by controlling atmospheric $CO_2$ level. Knowledge of gross primary production (GPP) for forest ecosystems is indispensable for the estimation of forest carbon storage. GPP is the first entry of atmospheric carbon into the forest ecosystem via photosynthesis. Process-based forest simulators (PBS) evaluate forest ecosystem activity by simulating different physiological plant responses to climatic conditions, atmospheric properties and plant structures (Constable and Friend, 2000; Running, 1994).

Simulating PBS requires input parameters that distinguish different vegetation types by their physiological and morphological characteristics. Implementation of a PBS for specific sites is complicated by the large number of parameters for plants, the soil and the atmosphere. Field measurements of PBS parameters are difficult or impossible to obtain, leading to incomplete knowledge of site specific parameters for the occurring species. In practice practitioners often rely on the literature for values of the PBS parameters (Hartig et al., 2012; Mäkelä et al., 2000).

A systematic adjustment of PBS parameters is required within the margins of the uncertainty so that the simulated outputs (e.g., GPP) satisfy pre-agreed criteria. This adjustment of a simulator parameters is called calibration. Calibration is often performed to obtain single optimized values of the parameters without the quantification of uncertainty in the parameters and the simulated outputs. Quantification of uncertainty is important for both scientific and practical purposes (Hamm et al., 2015b; Verstegen et al., 2014; He et al., 2014; Bastin et al., 2013).

A Bayesian framework provides a coherent method for calibrating a PBS (van Oijen et al., 2011; Reinds et al., 2008; van Oijen et al., 2005) and involves the identification of uncertainty in the parameters from the available information. This uncertainty is expressed as the prior probability distributions of the parameters. Independent observations of the variables corresponding to the PBS outputs (e.g., GPP) are used to update the prior probability distributions by means of Bayes rule. This updating generates the posterior probability distributions of the parameters, which can be summarized as medians and 95% credible intervals as the quantification of uncertainty. Hence a Bayesian framework combines prior probability distributions of the parameters and the observations to quantify uncertainty in the parameters and the PBS outputs.

In this study, a widely used simulator BIOME-BGC (Thornton, 1998) was calibrated in a Bayesian framework for a single output variable, GPP. A systematic search of the literature was used to construct the prior probability distributions on the BIOME-BGC parameters (Raj et al., 2014). A time series of daily flux tower GPP, partitioned from the flux tower measurements of net ecosystem exchange (NEE), provided independent GPP observations (Raj et al., 2016). We used flux tower GPP to update the priors of BIOME-BGC parameters. In principle, NEE data could be used alone to calibrate BIOME-BGC, where NEE is derived as the difference between the GPP and ecosystem respiration ($R_{eco}$). Hence, a calibration of BIOME-BGC using NEE data only ensures the accuracy of difference between GPP and $R_{eco}$ (Mitchell et al., 2011). The accuracy of simulated GPP can not be achieved using the NEE data alone. Our study focused on achieving the accuracy of simulated first entry of atmospheric carbon, i.e., GPP, into the ecosystem. Therefore we used partitioned flux tower GPP data to calibrate BIOME-BGC.

BIOME-BGC simulates GPP at daily time-step and it updates its memory between days (Thornton, 2010; White et al., 2000). This memory corresponds to the mass (amount of carbon) stored in different components of the vegetation, litter, and soil. The update of memory is directly related to the seasonal development of the state variables such as carboxylation capacity ($V_{cmax}$) that in turn controls the seasonality of simulated GPP. Input parameters are important to control the seasonality of the state variables and thus of GPP. BIOME-BGC accepts constant values of the input parameters during a simulation over the entire study period. We hypothesized that the seasonal cycle of GPP was not accurately captured by the constant (time-invariant) parameters and that their temporal variations could probably improve the seasonal cycle of GPP. We, therefore, further investigated if the temporal variation in the input parameters could be captured by means of Bayesian calibration.

The objective of this study was to quantify the uncertainty in BIOME-BGC input parameters and simulated GPP by integrating flux tower GPP into BIOME-BGC in a Bayesian framework. We obtained the posterior BIOME-BGC parameters: a) by calibrating the BIOME-BGC to the data of entire study period (growing season); and b) by calibrating the BIOME-BGC to 1 month of GPP data, and repeating the calibration for all months in the growing season. The main novelty of this paper is the presentation of a Bayesian framework for BIOME-BGC parameters estimation. The simulator itself is left unaltered. Additionally, investigation of temporal variation in BIOME-BGC input parameters would also reinforce to reconsider the assumption of constant parameters of other process-based simulators for photosynthesis.

## 2   Site description

Calibration of BIOME-BGC was performed at the Speulderbos forest site, which is located at $52^o15'08''$ N, $05^o41'25''$ E within a large forested area in the Netherlands. There is a flux tower within a dense 2.5 ha Douglas fir stand, which is a type of evergreen needleleaf species. The stand was planted in 1962. The vegetation, soil, and climate of this site have been thoroughly described elsewhere (Raj et al., 2014; Su et al., 2009; van Wijk et al., 2001; Steingrover and Jans, 1994).

## 3   Methods

### 3.1   Data and simulators

#### 3.1.1   The BIOME-BGC simulator

BIOME-BGC simulates biogeochemical processes including carbon, water and nitrogen fluxes within the vegetation, litter and soil compartment of terrestrial ecosystem at a daily time steps (Thornton et al., 2002; Running and Hunt, 1993). Evapotranspiration (ET), photosynthesis and respiration (autotrophic and heterotrophic) are the main processes simulated by BIOME-BGC. Simulation of daily ET is based on the Penman-Monteith equation (Monteith and Unsworth, 2008; McNaughton and Jarvis, 1983), which simulates ET as a function of incoming radiation, vapour pressure deficit (VPD) and the conductance associated with the evaporative surface. The photosynthetic routine uses Farquhar's biochemical model to estimate GPP (Thornton et al., 2002; Farquhar et al., 1980), which is the overall fixation of carbon. GPP is estimated independently for the sunlit and shaded canopy fractions. Final GPP is the sum of the contributions of the sunlit and shaded canopy fractions. GPP is a function of temperature, vapour pressure deficit, soil water content, solar radiation, atmospheric $CO_2$ concentration, leaf area index and leaf nitrogen concentration (Churkina and Running, 1998). The photosynthesis routine adds carbon to the system, which is removed from the system through respiration. A respiration routine computes autotrophic respiration as the sum of maintenance and growth respiration. Maintenance respiration is calculated as a function of leaf and root nitrogen concentration and tissue temperature. Growth respiration is the proportion of total new carbon allocated to growth. Heterotrophic respiration is the release of carbon through the process of decomposition of both litter and soil.

BIOME-BGC requires site characteristics, daily meteorological data, and ecophysiological parameters as inputs. The site characteristics include soil texture (percentage of sand, silt, and clay), elevation, latitude, shortwave albedo, wet and dry atmospheric deposition of nitrogen, symbiotic and asymbiotic fixation of nitrogen, and the effective soil rooting depth. We took the site characteristics data at Speulderbos from Raj et al. (2014). The meteorological data include daily minimum temperature (Tmin), daily maximum temperature (Tmax), the average daytime temperature (Tday), daily total precipitation, the daylight average shortwave radiant flux density (srad), the daylight average vapour pressure deficit (VPD) and the daylength from sunrise to sunset. We collected half-hourly temperature, prcp, srad, and relative humidity (RH) for each day in 2009 from the Speulderbos flux tower and daily values were obtained by the half hourly measurements. We derived VPD from relative humidity (RH) using the procedure described in Monteith and Unsworth (1990). BIOME-BGC requires 35 ecophysiological parameters for evergreen needleleaf forest/species (Table 1) and we obtained the prior uncertainty (expressed as a probability distribution) in each parameter for Speulderbos from Raj et al. (2014).

In this study, initial states of water, carbon, and nitrogen variables of the BIOME-BGC were prescribed with very low value ($\approx 0$) as recommended in Thornton and Running (2002); Thornton et al. (2002). Spin up simulation of BIOME-BGC was performed first to achieve steady state condition of soil carbon and nitrogen pools under given climate and site condition. Normal simulation was then started with these steady state condition using daily meteorological data of 2009.

### 3.1.2 Flux tower GPP data

We used observed data of net ecosystem exchange NEE to predict GPP at Speulderbos for the growing season (April to October) of 2009. To predict GPP, half-hourly GPP were separated from flux tower measurements of half-hourly net ecosystem exchange at Speulderbos site using non-rectangular hyperbola (NRH) model (Gilmanov et al., 2003). The estimation of the NRH model parameters was performed in a Bayesian framework that yielded posterior distributions of the NRH parameters and posterior predictions of GPP and its associated uncertainty (see Raj et al., 2016, for details). NEE was measured every half hour, leading to half-hourly predictions of GPP. These half-hourly values were summed to yield daily values of GPP (hereafter, referred as flux tower GPP) and its associated uncertainty (2.5 percentiles, 97.5 percentiles, and medians). Posterior distribution of NRH parameters were obtained for every 10-day blocks in the growing season (Raj et al., 2016). Since the parameters may vary over time for example due to dependencies on the factors that are not included directly in the NRH model (e.g., soil moisture, canopy structure, and nutrient limitations). Hence, although these factors (that affect GPP) are not included in the model, they are accounted for implicitly by the calibration to 10-day blocks of data.

### 3.2 Bayesian modelling

#### 3.2.1 Bayes rule

Bayesian calibration begins with Bayes rule (Gelman et al., 2013):

$$p(\boldsymbol{\theta}|\mathbf{z}) = \frac{p(\mathbf{z}|\boldsymbol{\theta})p(\boldsymbol{\theta})}{p(\mathbf{z})} \propto \text{likelihood} \times \text{prior,} \tag{1}$$

where $p(\boldsymbol{\theta})$ is the prior probability density function (pdf) of the parameters, in our study the BIOME-BGC parameters (e.g., FLNR, soil depth), contained in the vector $\boldsymbol{\theta}$. The term $p(\mathbf{z}|\boldsymbol{\theta})$ is the likelihood function, i.e., the conditional probability of observing the data $\mathbf{z}$ given $\boldsymbol{\theta}$. In our study, the vector $\mathbf{z}$ contains the independent observations of flux tower GPP, separated from NEE (see Section 3.1.2). The term $p(\mathbf{z})$ is the normalization constant independent of $\boldsymbol{\theta}$ and the term $p(\boldsymbol{\theta}|\mathbf{z})$ is the posterior pdf of $\boldsymbol{\theta}$ given the observed data.

The likelihood function is determined by the probability distribution of the residuals $\mathbf{e} = (e_1, e_2, ..., e_n)$, which are the difference between $\mathbf{y} = (y_1, y_2, .., y_n)$ and $\mathbf{z} = (z_1, z_2, .., z_n)$:

$$e_t = z_t - y_t, \quad t = 1, 2, ..., n \tag{2}$$

where, in our study, $\mathbf{y}$ denotes the simulated GPP (i.e., the output from the PBS). The residuals include the observation error and the simulator inadequacy, which arises due to the fact that the simulated output does not represent the true value of the

process even if $\boldsymbol{\theta}$ are known with no uncertainty (Kennedy and O'Hagan, 2001).

The posterior pdf in Eq. (1) cannot be obtained analytically for most practical problems. Inference is performed using the unnormalized density (Gelman et al., 2013) using Markov chain Monte Carlo (MCMC) simulation (Vrugt, 2016; Gelman et al., 2013; Vrugt et al., 2009; Gelfand and Smith, 1990; Hastings, 1970; Metropolis et al., 1953), as described in Section 3.2.2.

#### 3.2.2 DiffeRential Evolution Adaptive Metropolis (DREAM)

We adopted the DREAM algorithm proposed by Vrugt et al. (2009, 2008) to implement MCMC. DREAM stands for DiffeRential Evolution Adaptive Metropolis. DREAM runs $N$ different Markov chains in parallel for each $\theta_j$. Let the vector of simulator parameters $\boldsymbol{\theta} = (\theta_1, \theta_2, \theta_3, ...., \theta_d)$. The current state of the $i^{th}$ chain is given by single $d$-dimensional parameter vector $\boldsymbol{\theta}^{(i)}$. The $N$ Markov chains make $N$ such vectors $\boldsymbol{\theta}^{(1)}, \boldsymbol{\theta}^{(2)}, ....., \boldsymbol{\theta}^{(N)}$. The following steps explain briefly the DREAM algorithms.

1. For each chain $i$ ($i = 1, 2, ..., N$), an arbitrary starting point $\boldsymbol{\theta}^{(i)}$ from the prior pdf of the parameters are sampled.

2. A simulator is run at the starting points and and the likelihood $p(\mathbf{z}|\boldsymbol{\theta}^{(i)})$ ($i = 1, 2, ..., N$) is obtained. The density $p(\boldsymbol{\theta}^{(i)}|\mathbf{z})$ is then obtained for each chain:

$$\begin{aligned} p(\boldsymbol{\theta}^{(i)}|\mathbf{z}) &= p(\boldsymbol{\theta}^{(i)}) \times p(\mathbf{z}|\boldsymbol{\theta}^{(i)}) \\ &= \left\{ p(\theta_1^{(i)}) \cdot p(\theta_2^{(i)}) \cdot ..... \cdot p(\theta_d^{(i)}) \right\} \times p(\mathbf{z}|\boldsymbol{\theta}^{(i)}). \end{aligned} \tag{3}$$

The choice of likelihood and prior pdf of $\boldsymbol{\theta}$ for BIOME-BGC are explained in Sections 3.3.2 and 3.3.1 respectively.

3. For $i = 1,2,3,.....,N$:

   (a) A candidate point $\boldsymbol{\theta}^{(i)*}$ in chain $i$ is generated from the randomly chosen pairs of chains:

$$\boldsymbol{\theta}^{(i)*} = \boldsymbol{\theta}^{(i)} + (\mathbf{1}_d + \boldsymbol{\lambda}_d)\gamma(\delta, d)\left(\sum_{k=1}^{\delta}\boldsymbol{\theta}^{(k)} - \sum_{l=1}^{\delta}\boldsymbol{\theta}^{(l)}\right) + \boldsymbol{\zeta}_d,$$  (4)

   and

$$\gamma = 2.38/\sqrt{2\delta d}.$$

   where $\delta$ is the number of chain pairs used to generate the candidate point, $\boldsymbol{\theta}^{(k)}$ and $\boldsymbol{\theta}^{(l)}$ are randomly selected from the state of other chains; $k, l \in (1, 2, .., N)$ and $k \neq l \neq i$. The values of $\boldsymbol{\lambda}_d$ and $\boldsymbol{\zeta}_d$ are sampled from the uniform distribution $U(-b, b)$ and the normal distribution $\mathcal{N}(0, c)$ respectively. The typical default values of $\delta = 3$, $b = 1$, and $c = 10^{-6}$. $\gamma$ is the jump-size, whose value depends on $\delta$ and $d$. DREAM implements a randomized subspace sampling, i.e., all dimensions of $\boldsymbol{\theta}^{(i)}$ are not updated jointly and some dimensions of $\boldsymbol{\theta}^{(i)*}$ are reset to those of $\boldsymbol{\theta}^{(i)}$. The value of $\gamma$ is, therefore, obtained with $d'$, the number of dimensions updated jointly.

   (b) The simulator is run at the candidate point $\boldsymbol{\theta}^{(i)*}$ and the density $p(\boldsymbol{\theta}^{(i)*}|\mathbf{z}) = p(\boldsymbol{\theta}^{(i)*}) \times p(\mathbf{z}|\boldsymbol{\theta}^{(i)*})$.

   (c) The Metropolis ratio is given as $p(\boldsymbol{\theta}^{(i)*}|\mathbf{z}) / p(\boldsymbol{\theta}^{(i)}|\mathbf{z})$

   (d) The candidate point $\boldsymbol{\theta}^{(i)*}$ is accepted if the Metropolis ratio is larger than an acceptance criterion, which is a random number generated from the uniform distribution between 0 and 1. This may allow acceptance of $\boldsymbol{\theta}^{(i)*}$ with a lower likelihood than the current candidate point.

   (e) If the candidate point is accepted: $\boldsymbol{\theta}^{(i)} = \boldsymbol{\theta}^{(i)*}$, otherwise it remains at $\boldsymbol{\theta}^{(i)}$.

4. All $N$ Markov chains evolve in parallel for $T$ times by repeating Step 3. In order to perform inference using the Markov chains it is important that the chains have converged to a stationary distribution that is independent of their initial values. This is evaluated using diagnostic statistics and diagnostic plots, as described in Section 3.3.3. Unconverged chains are discarded as "burn-in" and the post burn-in samples are then used to conduct inference on each $\theta_j$. The post burn-in samples are then used to conduct inference on each $\theta_j$. For example median and 95 % credible interval can be obtained over these samples. A simulator is run on the posterior distributions of $\boldsymbol{\theta}$ to get the uncertainty in the simulated output (e.g., GPP for BIOME-BGC).

   The choice of $N$, $T$, and burn-in period are discussed in Section 3.3.3. The convergence diagnostic of Markov chains are also explained further in Section 3.3.3.

## 3.3 Implementation of DREAM for BIOME-BGC

### 3.3.1 Prior distributions of the BIOME-BGC parameters

The computational load of Bayesian calibration of a simulator can be reduced by excluding those input parameters that have negligible influence on the simulated output (Minunno et al., 2013; van Oijen et al., 2013; Xenakis et al., 2008). BIOME-BGC

requires 35 ecophysiological parameters for evergreen needleleaf species (Table 1), each having a varying degree of influence on the simulated GPP. Raj et al. (2014) conducted variance-based sensitivity analysis (VBSA) of BIOME-BGC at Speulderbos to investigate the sensitivity of simulated GPP to the ecophysiological parameters and the effective soil rooting depth. They treated soil rooting depth as a parameter. For VBSA, they identified the uncertainty in each ecophysiological parameter and the soil depth in the form of pdfs. They found that GPP is mainly sensitive to 5 ecophysiological parameters and the soil rooting depth, while others were found to have negligible influence on simulated GPP. In this study, we included these 6 input parameters (highlighted in Table 1) for calibration, whose prior pdfs were assumed identical to that identified by Raj et al. (2014). Other input parameters were fixed at the mean value of the distribution provided by Raj et al. (2014).

[Table 1 about here.]

### 3.3.2 The likelihood

Recall from Section 3.2.1 that the likelihood is determined by the pdf of the residuals, $e_t = z_t - y_t$ (Eq. 2 ). Hence, the likelihood function evaluates how well the BIOME-BGC simulated GPP, $\mathbf{y}$, is able to reproduce the data, $\mathbf{z}$. The likelihood function typically defined assuming that the residuals are independent and identically normally distributed (Starrfelt and Kaste, 2014; Braakhekke et al., 2013; Reinds et al., 2008; Svensson et al., 2008; van Oijen et al., 2005). This assumes that the simulator models perfectly the temporal profile of GPP leaving no residual temporal correlation in the residuals from the time series. This assumption may not be correct.

BIOME-BGC simulates the time series of GPP at daily time steps. We relaxed the assumption that the temporal profile of simulated GPP perfectly follows the flux tower GPP and modelled the temporal correlation in the residuals. We adopted a likelihood that assumes the residuals follow an autoregressive process of order one (Vrugt, 2016), given as:

$$
p_{\log}(\mathbf{z}|\boldsymbol{\theta}) = -\frac{n}{2}\log(2\pi) + \frac{1}{2}\log\left(1 - \phi^2\right) - \frac{1}{2}\left(1 - \phi^2\right)\hat{\sigma}_1^{-2}e_1^2 - \sum_{t=2}^{n}\log(\hat{\sigma}_t)
$$
$$
-\frac{1}{2}\sum_{t=2}^{n}\left(\frac{e_t - \phi e_{t-1}}{\hat{\sigma}_t}\right)^2. \tag{5}
$$

where $\phi$ and $\hat{\sigma}$ are nuisance parameters that are inferred jointly with $\boldsymbol{\theta}$. The parameter $|\phi| < 1$ accounts for the temporal correlation in the residuals, $\mathbf{e}$ and $\phi = 0$ means that there is no temporal correlation. We evaluated whether the posterior distribution $\phi$ were different from zero (Section 4.1). A uniform prior distribution of $\phi$ between -1 and +1 was chosen as recommended in Vrugt (2016).

Equation 5 gives the likelihood on the logarithmic scale. This improves numerical stability by avoiding rounding errors in the computation. $n$ is the length of the vectors $\mathbf{z}$ and $\mathbf{y}$.

If the error residuals are assumed to be uncorrelated, Eq. 5 reduces to the following equation:

$$
p_{\log}(\mathbf{z}|\boldsymbol{\theta}) = -\frac{n}{2}\log(2\pi) - \sum_{t=1}^{n}\log(\hat{\sigma}_t) - \frac{1}{2}\sum_{t=1}^{n}\left(\frac{e_t}{\hat{\sigma}_t}\right)^2. \tag{6}
$$

We also checked the changes in the results using the likelihood function not accounting for correlation in the residuals (Eq. 6). Mainly, the results, given below, were obtained using the likelihood function with temporal correlation in the residuals (Eq. 5). Whenever, we have presented the results using the likelihood function given by Eq. 6, we have specifically mentioned this.

### 3.3.3 Posterior prediction of BIOME-BGC parameters and GPP

We implemented the DREAM algorithm in MATLAB version R2015b. The DREAM toolbox was provided by its developer, Jasper A. Vrugt, from University of California, Davis, USA. Technical details of the DREAM toolbox are provided by Vrugt (2016).

We used $N = 10$ Markov chains with $T = 15000$ iterations for each chain. This produced 150000 ($N \times T$) posterior samples for each $\theta_j$ ($j = 1, 2, .., 6$ for selected BIOME-BGC parameters for calibration). Gelman et al. (2013) and Vrugt et al. (2009) recommend discarding the first 50% of the samples as a burn-in; however, we discarded 10000 samples, in order to reduce the computation cost. This resulted in 50000 ($N \times (T - \text{burn-in})$) post burn-in samples for each $\theta_j$. The convergence of these post burn-in samples was evaluated using the Gelman–Rubin diagnostic (Gelman and Rubin, 1992) and through visual examination of the trace plots. The Gelman–Rubin potential scale reduction factor (PSRF) compares the between-chain and within-chain variance of the parallel Markov chains. A PSRF close to 1 indicates that the chains have converged.

The post burn-in samples created 50000 vectors of $\boldsymbol{\theta}$. BIOME-BGC was run at each parameter vector using daily meteorological data of 2009 and the daily simulated GPP (hereafter refereed as posterior GPP) was evaluated and stored. This produced the distribution of daily posterior GPP, which was summarized by the median and the 2.5 and 97.5 percentile (i.e., 95 % credible interval). The 95 % credible interval showed the uncertainty in the daily posterior GPP. We compared these 95 % credible intervals and medians over the growing season with that of flux tower GPP.

We conducted two experiments to obtain the posterior samples of $\boldsymbol{\theta}$:

1. Experiment 1: We used daily mean of flux tower GPP for five months in the growing season (April to August 2009) to calibrate BIOME-BGC for the growing season. For calculation of the likelihood using Eq. (5), we set $n = 153$, equal to the number of days in April to August. Note that we did not include the daily flux tower GPP for September and October in the calibration and we used these data for validation of the calibrated BIOME-BGC. In this experiment, the posterior samples of $\boldsymbol{\theta}$ were used to obtain posterior GPP and the associated uncertainty for each day in 2009. The procedure of Experiment 1, stated above, were also repeated using the likelihood function given by Eq. 6.

2. Experiment 2: We used daily mean of flux tower GPP for one month only, e.g., April, in the growing season to calibrate BIOME-BGC. For the calculation of likelihood using Eq. (5), we set $n = 30$, equal to the number of days in April. The posterior samples of $\boldsymbol{\theta}$ were used to obtain posterior GPP and the associated uncertainty for each day in 2009. We then extracted the daily posterior GPP (with the associated uncertainty) of April only and discarded the other months in 2009. Likewise, we obtained posterior GPP and the associated uncertainty for the other six months (May to October 2009) in the growing season. Experiment 2 resulted in seven different posterior samples of $\boldsymbol{\theta}$.

For both experiments, we followed the same procedure explained in the paragraph 2 and 3 of this section.

### 3.3.4 Statistical evaluation of BIOME-BGC simulated GPP

We determined the performance of the calibration using two criteria that evaluate efficiency with which the calibrated BIOME-BGC reproduces the flux tower GPP. Both criteria provide a single measure of BIOME-BGC efficiency in simulating daily GPP over the selected period. The first criterion was the root mean square error (RMSE) between the simulated and flux tower GPP:

$$\text{RMSE} = \sqrt{\frac{1}{n} \sum_{t=1}^{n} (z_t - y_t)^2}, \tag{7}$$

where $n$ is the number of daily flux tower GPP ($z_t$) and the simulated GPP ($y_t$). RMSE has the unit of GPP. A low value of RMSE indicates high accuracy. The second criterion was the Nash-Sutcliffe efficiency (NSE) (Nash and Sutcliffe, 1970):

$$\text{NSE} = 1 - \frac{\sum_{t=1}^{n} (z_t - y_t)^2}{\sum_{t=1}^{n} (z_t - \overline{z})^2}, \tag{8}$$

where $\overline{z}$ is the mean of the observations (flux tower GPP). NSE can range from $-\infty$ to $1$. An NSE value close to $1$ indicates high accuracy in the simulation of GPP. Following Dumont et al. (2014), we assumed that a NSE $\geq 0.5$ indicates adequate accuracy in the simulated GPP.

We evaluated the performance of BIOME-BGC for the following cases:

1. For Experiment 1, we obtained RMSE and NSE for the two periods: calibration period of five months (April to August) and the validation period of two months (September and October). For each period, the calculations were made for 2.5 percentiles, 97.5 percentiles, and medians. Note that the RMSE and NSE are typically evaluated at the median of the posterior predictive distribution; however, this does not evaluate the posterior uncertainty (Hamm et al., 2015a). Therefore we also calculated the RMSE and NSE for the 2.5 and 97.5 percentiles of the posterior GPP ($y_{2.5}$ and $y_{97.5}$) against the same percentiles of flux tower GPP ($z_{2.5}$ and $z_{97.5}$).

2. For Experiment 2, we obtained RMSE and NSE for the same two periods and percentiles as stated in point 1 (above), to make a direct comparison with the results of Experiment 1.

3. To show the performance of uncalibrated BIOME-BGC, we obtained the daily simulated GPP with 95% credible intervals at the prior distributions of six selected parameters (Table 1). We sampled from these prior distributions to obtain 50000 parameter vectors. BIOME-BGC was run at these parameter vectors to yield the prior predictor of BIOME-BGC simulated GPP (hereafter referred as prior GPP). We calculated the RMSE and NSE for the same two periods and percentiles as stated in point 1, to make a direct comparison with Experiments 1 and 2.

## 4 Results

### 4.1 Convergence of the Markov chains

The value of Gelman–Rubin PSRF was close to one for each $\theta_j$ obtained in both experiments (Table 2). Figures 1a to 1f show the trace plots of each $\theta_j$ for Experiment 1. Visual inspection of the trace plots indicated that all ten Markov chains were mixed properly with each other. For Experiment 2, we also observed the convergence of the Markov chains for each $\theta_j$ in each month of the growing season (trace plots not shown here). The visual and statistical diagnostic demonstrated that each $\theta_j$ had explored its range and the obtained samples from the converged chains were the samples from the posterior distribution.

Figure 1g shows the trace plot of $\phi$, accounting for the temporal correlation in the error residuals (section 3.3.2), for Experiment 1. We observed $\phi \neq 0$ and its value ranged from 0.56 to 0.93 with a mean at 0.75. The non-zero values of $\phi$ indicated that the residuals are temporarily correlated, thus supporting our choice of likelihood function (Eq. 5). For Experiment 2, non-zero values of $\phi$ were also obtained in each month.

[Figure 1 about here.]

[Table 2 about here.]

### 4.2 Posterior distribution of BIOME-BGC parameters

Figure 2 shows the temporal profile of median and 95% credible interval of each $\theta_j$ over the growing season for Experiment 2. For Experiment 1, we obtained a single value for the median and 95 % credible interval. For both experiments, we observed that the uncertainty in the posterior distribution of each $\theta_j$ was reduced compared to the prior distribution, indicating that $\boldsymbol{\theta}$ were constrained by the flux tower GPP observations. These uncertainties were higher in Experiment 2 than that of Experiment 1. The upper quantiles (97.5%) of the posterior distributions of the parameters LFRT, FRC:LC, and SD were found close to the maximum values of the corresponding prior distributions for both experiments. The uniform priors of these parameters (Table 1) possibly imposed an upper boundary in the posteriors, which is called edge effect. Prior uniform distributions could be made wider in order to eliminate the edge effect. But, we chose to keep these maximum values since the choices, given in Table 1, were based on the realistic ranges of LFRT, FRC:LC, and SD for Dougls-fir at Speulderbos. For FRC:LC, previous work (Raj et al., 2014) on the study area found the maximum limit of FRC:LC up to 6.85. But we did not use the limit of 6.85 to make the uniform distribution of FRC:LC wider in the present study. Raj et al. (2014) found that the increase of upper limit of the uniform distribution of FRC:LC from 2.16 to 6.85 led to the simulation with no development in LAI (leaf area index) and hence no production at the study site. The upper limit of FRC:LC at 2.16, however, fully supported the development of LAI at the study site. Therefore, we kept the upper limit of FRC:LC at 2.16 in the present study.

A Bayesian calibration also allowed us to obtain correlation between the calibrated parameters. Figure 3 shows the correlation coefficients "$r$" and scatterplots between the posterior distributions of two parameters, obtained in Experiment 1, of different pair combinations. A strong positive correlation was found between the posterior distributions of C:N$_{\text{leaf}}$ and FLNR with $r = 0.95$. This strong positive correlation is in-line with the formulation of FLNR that shows direct proportionality with

C:N$_{leaf}$ (see Appendix A in Raj et al., 2014, for details). The parameters C:N$_{leaf}$ and FLNR showed similar negative, but weak (> -0.5), correlation with $W_{int}$ ($r \approx -0.4$). This can be explained by the fact that the simulated GPP is expected to vary inversely with $W_{int}$ via soil water potential and stomatal regulation and directly with FLNR and C:N$_{leaf}$ (see Section 5.1, for details of BIOME-BGC internal routines). The parameter SD had similar positive, but weak (< 0.5), correlation with FLNR and C:N$_{leaf}$ ($r \approx 0.4$). This can be explained by the fact that the simulated GPP is expected to vary directly with SD (via soil water potential and stomatal regulation), and FLNR and C:N$_{leaf}$. Two parameters of any other pair combinations did not show any notable correlation.

For Experiment 2, the uncertainties in LFRT, FRC:LC, $W_{int}$, and SD were higher at the start and end of the growing season compared to other months. The uncertainties in these parameters were lowest for calibration to GPP values of the peak of the growing season (July and August). The values of LFRT, FRC:LC, and SD increased during the peak of the growing season and became close to that obtained in Experiment 1 and then started decreasing. The opposite trend was observed for $W_{int}$. The uncertainty in C:N$_{leaf}$ for any month obtained in Experiment 2 was comparable and within the range of that obtained in Experiment 1. We did not find significant variation in the trend of FLNR obtained in Experiment 2 during the growing season; however, higher uncertainty in FLNR was observed compared to Experiment 1.

[Figure 2 about here.]

[Figure 3 about here.]

## 4.3 Evaluation of calibrated BIOME-BGC for Experiment 1

We evaluated the performance of calibrated BIOME-BGC by comparing the daily posterior GPP and the daily flux tower GPP, for the calibration period of April to August (Fig. 4) and the validation period of September and October (Fig. 5). The daily posterior GPP were summarized by the median and 95% credible interval. The temporal profile of these medians and credible intervals were plotted against that of flux tower GPP. Evaluation of the BIOME-BGC before and after calibration (Experiment 1) based on the statistical criteria (RMSE and NSE) is shown in Table 3. The periods for which these criteria were obtained are explained in Section 3.3.4.

Overall, daily posterior GPP was close to flux tower GPP during the calibration period (Fig. 4), although the separation between these two temporal profiles in April (Julian day 91 to 120) was large compared to other months (Julian day 121 to 242) in the growing season. For the validation period, posterior GPP closely followed the flux tower GPP (Fig. 5).

The posterior GPP was improved compared with the prior GPP, as indicated by the drop of RMSE for the median as well as the 2.5 and 97.5 percentile for both calibration and validation periods (Table 3). The NSE criterion was also improved after calibration (NSE > 0.5), whereas before calibration, the value of NSE was negative. The enhancement in NSE and the drop of RMSE give statistical evidence of the improvement in the daily prior GPP after calibration.

We also evaluated the performance of calibrated BIOME-BGC using the likelihood function without the temporal correlation in the residuals (Eq. 6). The obtained daily medians of posterior GPP for the calibration period (April to August) are shown in Fig. 4. For daily medians as well as 2.5 and 97.5 percentile, RMSE and NSE criteria are shown in Table 3. We found that both

likelihood functions Eq. 5 and Eq. 6 led to similar temporal profile of the posterior GPP and similar values of RMSE and NSE criteria.

[Figure 4 about here.]

[Figure 5 about here.]

[Table 3 about here.]

## 4.4 Posterior GPP for Experiment 2

Combining the daily simulations of each month provided the temporal profile of the medians and 95% credible intervals of the daily posterior GPP over the growing season. Figure 6 shows this temporal profile (black line and grey shade) from April to August. We observed that the posterior GPP had a better fit to the flux tower GPP, compared to Experiment 1 (Fig. 4). Particularly, the posterior GPP of April (Julian days 91 to 120) followed the flux tower GPP more closely than Experiment 1. We found further enhancement in the NSE compared to Experiment 1 for the median, 2.5, and 9.5 percentile over the period of April to August (Table 3) where the values of NSE became closer to 1. A drop in RMSE was also observed. For the period of September and October (temporal profile not shown here), however, the NSE and RMSE were the same as for Experiment 1. These results indicated an improvement in the posterior GPP compared to that obtained from Experiment 1, but at the expense of a higher degree of freedom.

[Figure 6 about here.]

## 5 Discussion

### 5.1 Simulation of GPP using BIOME-BGC

To explain our results, we identified the processes within BIOME-BGC that are controlled by the six calibrated parameters and relate to the simulation of GPP (Fig. 7). These processes are implemented by different routines. The routines, however, are controlled not only by these six parameters, but also generate intermediate outputs, as shown in Fig. 7. We only highlight those routines that were relevant to the simulation of GPP. We refer the reader to Thornton (2010) for a detailed explanation of the routines.

BIOME-BGC simulates the daily development of plant carbon pools (White et al., 2000). The development of carbon pools is governed by the daily update of BIOME-BGC memory of mass of carbon stored in different components of the plant. The simulated development of plant carbon pools on a particular day is dependent on the previous days. BIOME-BGC converts the carbon stored in the leaf pool (leaf C) into an equivalent leaf area index (LAI). The development of leaf C controls the development of LAI in the radiation transfer routine. Leaf C relates to the loss of leaf biomass, which is expressed as the parameter LFRT. The parameter FRC:LC is also responsible for the development of leaf C and then LAI. In the precipitation

routine, $W_{int}$, together with LAI, determines the amount of precipitation intercepted by the canopy, which in turns controls the amount of water that reaches the soil. The soil psi routine calculates the volumetric water content in the soil as the ratio of soil water to SD. Thereafter, soil water potential is derived as a function of volumetric water content. The soil water potential acts as a multiplier in the evapotranspiration routine to simulate stomatal closure and the leaf scale conductance to water vapour per unit leaf area.

The photosynthesis routine converts the conductance to water vapour to the conductance for $CO_2$, which measures the rate of passage of $CO_2$ into the leaf stomata. The parameter $C:N_{leaf}$ together with LAI determines the leaf nitrogen content from the carbon pool in the photosynthesis routine and the day leaf maintenance respiration per unit leaf area in the respiration routine. The leaf scale conductance to $CO_2$, leaf nitrogen content, day leaf maintenance respiration and the parameter FLNR are further used in the Farquhar model, implemented in the photosynthesis routine, to simulate the carboxylation capacity ($V_{cmax}$) and thus the carbon assimilation. The assimilated carbon is then added to the day leaf maintenance respiration and then multiplied by the LAI and daylength to simulate the daily GPP. The respiration routine also calculates the maintenance respiration of roots and stems (not shown in Fig. 7) together with leaves. The respiration terms are summed and then subtracted from GPP to obtain available carbon for allocation, which further updates leaf C. Finally, Fig. 7 indicates which meteorological variables are used in a given routine, although we have not described their specific role.

We presented the link between six calibrated parameters and the BIOME-BGC internal routines so that we could explain our results considering the development of the state variables, principally such as LAI and $V_{cmax}$. LAI and $V_{cmax}$ exhibit a seasonal cycle and affect the seasonality of simulated GPP. This is explored further in Section 5.2.

[Figure 7 about here.]

## 5.2 BIOME-BGC calibration

BIOME-BGC accounts for dynamic in carbon stocks in the vegetation by means of allocation. Hence it uses parameters that are constant for the year of simulation. Consider Experiment 1. The memory of BIOME-BGC is updated between days (section 5.1), and Biome-BGC takes care of the simulation of time-varying state variables such as leaf area index (LAI) and carboxylation capacity ($V_{cmax}$) used in Farquhar's model. Therefore, the daily simulated GPP are temporarily dependent. The posterior GPP closely followed the flux tower GPP even for those months (September and October) which were not included in the calibration (Fig. 5), although this was not perfect as shown by the fact that $\phi \neq 0$. If the simulator would properly capture the temporal development of GPP we would expect that $\phi = 0$, even after allowing for some uncertainty in the prediction. We deliberately assumed $\phi = 0$ in the likelihood function (Eq. 6) to check if this assumption has any effect on the posterior GPP. We, however, found that both choices $\phi \neq 0$ and $\phi = 0$ led to similar posterior GPP (section 4.3). This comparison indicated that an improvement in temporal development of GPP after calibration might not be achieved, at least for BIOME-BGC simulator, with either the assumption of presence or absence of temporal correlation in the residuals. The representation of dynamic processes within the simulator responsible for GPP should be, therefore, given more attention in order to improve the temporal development of GPP. This is what we showed in Experiment 2.

Experiment 1 showed that BIOME-BGC was able to reproduce closely the flux tower GPP. Further, the Bayesian calibration allowed daily posterior GPP simulation as well as quantification of the associated uncertainty (Figs. 4 and 5). The edge effect in the posterior distributions of the parameters LFRT, FRC:LC, and SD (Section 4.2) could be seen as the deficiency of the calibration. A drop in RMSE and enhancement in NSE coefficient (Table 3) before and after calibration, however, indicated the efficiency of the calibration. Furthermore, the apparent overprediction of daily posterior GPP, compared to flux tower GPP, for the month of April raised questions: (a) on the reliability of posterior GPP for those months that were not included in this study; and (b) whether the seasonal cycle of all of the state variables was simulated realistically. These questions led us to estimate the posterior distributions of parameters for different months representing the phenological cycle in Experiment 2.

Consider Experiment 2. Note that BIOME-BGC actually simulated daily posterior GPP for a whole year with the posterior distributions of the parameters of each month. We selected only the daily posterior GPP of that month to which the posterior distributions belong and we discarded the other eleven months of simulations. The temporal profile in Fig. 6 is the combinations of daily posterior GPP of each month in the growing season (Section 3.3.3 and 4.4). Thus the temporal profile of daily posterior GPP in Experiment 2 was obtained by mixing several independently simulated time series. The resulting time series has discontinuities in state variables and thus the update of simulator memory (section 5.1) between the months is ignored. This time series can, however, help to analyse the simulator behaviour for the temporal variation in the input parameters. Alternatively, one could think of updating the simulator state at the end of a month. This would then be the starting state for the run of the next month with changed parameters. This approach, however, can not be implemented in the original configuration of BIOME-BGC, because a single forward run of BIOME-BGC simulates output for at least one year and accepts only constant input parameters. These parameters can not be changed across months in a single forward run. This would require changing the BIOME-BGC code. Such a modification was, however, not desired because model deficiency of BIOME-BGC could still be investigated through the temporal variation in the input parameters across the season using the approach proposed in Experiment 2. BIOME-BGC was therefore calibrated against the data of each month separately, as if information on GPP for the other months was absent. If the obtained variations in the input parameters improve the seasonality in simulated GPP, this indicates that the default linkage of the constant parameters with the state variables, that change during the season, in the simulator may require improvement in future study.

We observed an improvement (Fig. 6), particularly in the month of April, in the daily posterior GPP compared to Experiment 1 (Fig. 4). This improvement was also clear in Table 3 which shows an increase in the NSE and decrease in the RMSE for Experiment 2 compared to Experiment 1. More interestingly, Experiment 2 showed variation in the six calibrated parameters depending on the month the BIOME-BGC was calibrated to (Fig. 2), particularly $W_{int}$, SD , FRC:LC, and LFRT. These variations were also in-line with the seasonal variation in GPP. For example, maintaining the high GPP rates during the peak of the growing season (July and August), required lower $W_{int}$ and higher SD, both increasing the soil water availability through the precipitation routing routine in Fig. 7. During the start of growing season (April), higher $W_{int}$ and lower SD maintained low GPP rates. This suggests that either the soil water reservoir or the feedback mechanism between soil moisture and stomatal conductivity via the soil water potential was responsible for the under and overestimate of simulated GPP in Experiment 1. The parameters FRC:LC and LFRT were also higher when calibrated to summer months. Both parameters affect GPP through

LAI. The variation in FLNR and C:N$_{\text{leaf}}$, which together determine $V_{cmax}$, also changed month-by-month (Fig. 7). The results of Experiment 2 indicated that BIOME-BGC may be too rigid to simulate the seasonality of the state variables (LAI and $V_{cmax}$), at least in evergreen coniferous forest, without the temporal variation in the input parameters and thus highlighted the model deficiency of BIOME-BGC. To our knowledge, this aspect has not been discussed in earlier work on the calibration of

BIOME-BGC (Yan et al., 2014; Ueyama et al., 2010; Maselli et al., 2008).

The previous studies have also highlighted the improvement in the performance of simulator BEPS (Boreal Ecosystem Productivity Simulator) (Mo et al., 2008) and ORCHIDEE (ORganizing Carbon and Hydrology In Dynamic EcosystEms) (Williams et al., 2009) with varying the input parameters over time. Those studies provided insight to the poorly understood dynamical processes related to photosynthetic capacity. In our study, we re-examined the variation in the input parameters, related to pho-

tosynthetic capacity, of BIOME-BGC in a Bayesian framework. We observed that the temporal dynamics of the state variables (LAI and $V_{cmax}$) and the soil water mechanism within BIOME-BGC, and thus photosynthesis, are not sufficiently expressed by the constant input parameters. These state variables also control photosynthesis simulations in other process-based simulators, such as SCOPE (van der Tol et al., 2009), and are governed by the constant input parameters that may not be adequate based on our findings. Our study, therefore, reinforce a message that the reconsideration of temporal dynamics of state vari-

ables within the simulator, possibly through the temporal variation in the parameters, should receive further attention to the modelling communities focusing on simulating forest carbon cycle.

A major metrics of carbon cycle includes GPP, ecosystem respiration ($R_{eco}$) and NEE. In this study, we limited the calibration to partitioned flux tower GPP. A limitation of this approach is that the output of process-based simulators is calibrated against the output of another model, notably the flux partitioning model. The latter is not a process model, but a semi-empirical model

calibrated to 10-day's blocks of data (section 3.1.2). Although this approach has been used in many other studies that validate the output of process-based simulators (Zhou et al., 2016; Collalti et al., 2016; Liu et al., 2014; Yuan et al., 2014), it would also be possible to calibrate BIOME-BGC using this approach. More importantly, a calibration to NEE data (i.e., difference between GPP and $R_{eco}$) alone does not guarantee that GPP and $R_{eco}$ terms are well calibrated. Other studies (Kuppel et al., 2012; Fox et al., 2009) that used NEE data to calibrate process-based simulators DALEC (Data Assimilation Linked Ecosystem Carbon)

and ORCHIDEE , therefore, were more successful in achieving the accuracy of this simulated difference compared to GPP and $R_{eco}$. Tang and Zhuang (2009) showed the improvement (by the drop of RMSE) in simulated GPP and $R_{eco}$ by the process based simulator TEM (Terrestrial Ecosystem Model) when both GPP and $R_{eco}$ data were used in calibration as compared to using NEE data alone. In our study, we decided to test the calibration algorithm for GPP first. This approach can be extended to include $R_{eco}$ data together with NEE data in order to ensure the accuracy of all simulated metrics of carbon cycle. Then the

parameters, which may influence the simulated $R_{eco}$, need to be identified and should be included in the calibration.

We performed our calibration based on six parameters (LFRT, FRC:LC, C:N$_{\text{leaf}}$, $W_{\text{int}}$, FLNR, and SD) whereas BIOME-BGC has 35 parameters in total. A calibration based on 35 parameters was not feasible computationally so, in line with other authors (e.g., Minunno et al., 2013), we chose a subset of the parameters. We defend our choice of parameters based on our previous experimental results, which showed that annual total GPP was most sensitive to these parameters (Raj et al., 2014) at

Speulderbos. Nevertheless, GPP may be sensitive to other parameters at finer spatial scales. Computational developments and

the flexibility of the DREAM algorithm may allow more parameters to be calibrated. This could lead to a more comprehensive calibration to multiple outputs in the near future.

## 6 Conclusions

This study presented a Bayesian calibration framework for the simulator BIOME-BGC. We illustrated the framework at the Speulderbos forest site, the Netherlands. Use of the framework led to the following conclusions:

1. The Bayesian framework allowed quantification of uncertainty in both the estimated parameters and the posterior (predictive) GPP, through the posterior (predictive) distribution. The uncertainty is important in the sense that it helps to determine how much confidence can be placed in the results of forest carbon related studies based on GPP. A calibration based on optimization of BIOME-BGC parameters, as done in earlier studies, can not capture the associated uncertainty in the simulated GPP.

2. We modelled the temporal correlation in the residuals through the nuisance parameter, $\phi$, in the likelihood function. We concluded that BIOME-BGC did not properly simulate the temporal development of GPP, neither by assuming temporal correlation in the residuals ($\phi \neq 0$) nor by ignoring this ($\phi = 0$) and the dynamical processes within the BIOME-BGC became more prominent. Hence calibration gave greater insight into the simulator. Other future studies on the calibration of similar process-based simulators may also ignore $\phi$, but they should consider carefully the dynamic processes within the simulators to achieve improved calibration results.

3. We used the calibration results to gain further insights into the functioning (dynamic processes) of BIOME-BGC through analysis of the monthly variation in posterior parameter distributions. Our study revealed the model deficiency of BIOME-BGC for using constant parameters to simulate seasonality of state variables, thus the seasonality in daily GPP. The seasonality was captured more precisely by using monthly variation in the BIOME-BGC parameters. In future, such model deficiency should receive attention by the BIOME-BGC modelling communities. Nevertheless, our findings also suggest that the other modelling communities that use the similar process-based simulators may also consider to improve such model deficiency.

4. We implemented our calibration using the DREAM algorithm. DREAM offers considerable computational advantages and flexibility as compared to other MCMC implementations. It shows promise for biogeochemical and other environmental simulation applications. Specifically future research could calibrate more parameters.

## 7 Code and data availability

We provide a MATLAB script and input data as a supplementary material to support the implementation of Bayesian calibration of the BIOME-BGC simulator. The MATLAB script uses the functionality of the DREAM toolbox, which can be obtained, on request, from its developer Jasper A. Vrugt, University of California, Davis, USA (Vrugt, 2016). The source code of the

BIOME-BGC simulator can be downloaded from http://www.ntsg.umt.edu/project/biome-bgc. Markov chains in DREAM are run in parallel using multiple cores of the computer processors. DREAM consumes a large amount of memory (RAM). The experiment shown in this manuscript was performed on Windows Server 2012 Dell Precision 7910 with a twelve-core Xeon processor and 128 GB of RAM.

The description of each file in the supplementary material is given below:

1. MATLAB scripts

    (a) *DREAM_setup.m*: This scripts defines the basic settings of DREAM, which were used in our experiment. The script is self explanatory. This script calls the MATLAB function ("BIOME-BGCrunScript.m") to run BIOME-BGC simulator.

(b) *BIOME-BGCrunScript.m*: This scripts defines the function to run BIOME-BGC (by calling pointbgc.exe) simulator with the parameters value obtained in each iteration of DREAM and the daily simulated GPP (gross primary production) is returned. We do not provide "pointbgc.exe". This can be obtained by compiling BIOME-BGC source code.

2. Input data files used to run BIOME-BGC in our experiment (For details, see BIOME-BGC user guide that comes with

the source code of BIOME-BGC)

    (a) *enf_speuld_Main.ini*: This is the input initialization file. It provides general information about the simulation such as site characteristic data, the name of all required input files and output files, and lists of output variables to be stored.

    (b) *Meanpm.epc*: This is the input parameters file that contains mean value of each parameter.

(c) *Speuld2009.mtc41*: This file contains daily input meteorological variables of 2009 at Speulderbos site, the Netherlands.

3. Input flux tower GPP (for calibration and comparison with posterior simulated GPP)

    (a) *Percentiles_FluxTower_GPP_JD_91_304.xlsx*: This excel file contains mean and percentiles of daily GPP (for the growing season of 2009) partitioned from flux tower measurements of net ecosystem exchange at the Speulderbos

forest site, the Netherlands. Daily mean values were used in calibration and percentiles values were used to compare with posterior simulated GPP.

    (b) *TowerGPP.txt*: This file contains the subset (from Julian day 91 to 243 in Experiment 1, see Section 3.3.3) of daily mean of flux tower GPP. This file is called in "DREAM_setup.m". For Experiment 2, different subsets of flux tower GPP can be easily obtained from the file "Percentiles_FluxTower_GPP_JD_91_304.xlsx".

*Acknowledgements.* Biome-BGC version 4.2 was provided by Peter Thornton at the National Center for Atmospheric Research (NCAR), and by the Numerical Terradynamic Simulation Group (NTSG) at the University of Montana, USA. NCAR is sponsored by the National Science Foundation. The authors thankfully acknowledge the support of the Erasmus Mundus mobility grant and the University of Twente for funding this research. The authors acknowledge Jasper A. Vrugt, from University of California, Davis, USA for providing DREAM toolbox.

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

**List of Figures**

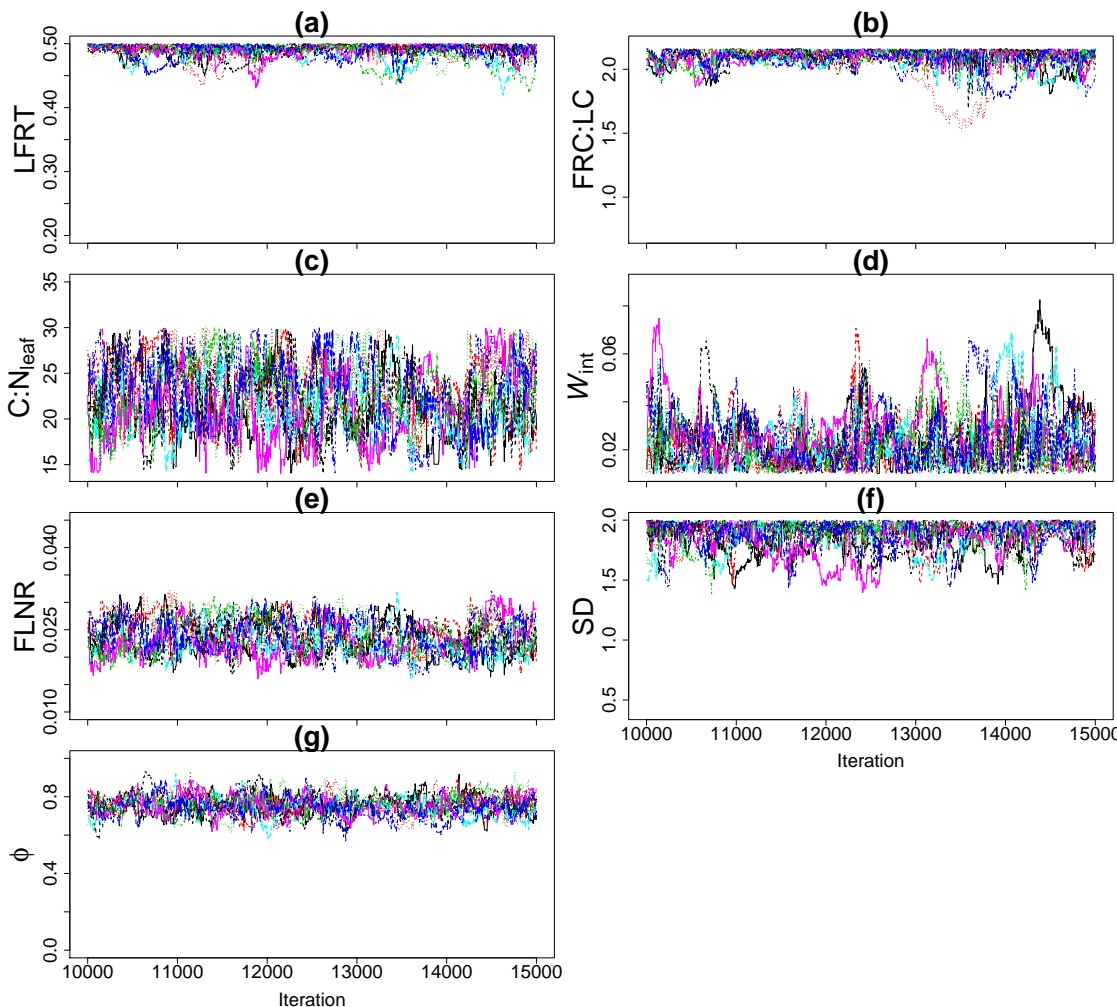

**Figure 1.** Trace plot of each calibrated BIOME-BGC parameter and $\phi$ (nuisance parameter of likelihood function Eq. 5) for Experiment 1 after burn-in period of 10000. Information about the BIOME-BGC parameters is given in Table 1. SD is effective soil depth.

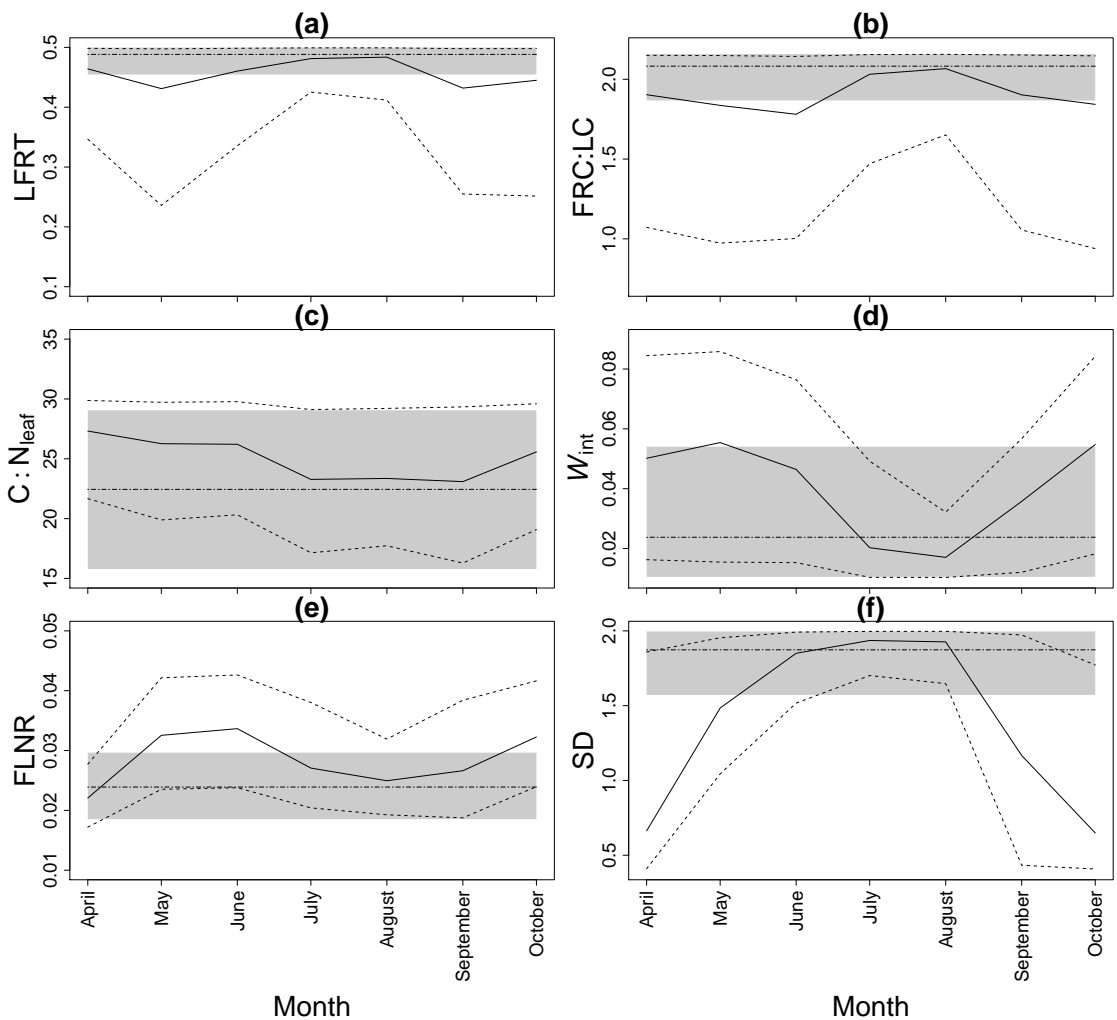

**Figure 2.** Median (solid lines) and 95% credible intervals (dashed lines) of the posterior distributions of each calibrated BIOME-BGC parameter obtained from Experiment 2 for each month during the growing season of 2009. The grey shade and dotted-dashed line represent median and 95% credible intervals obtained for Experiment 1. The range of the y-axis represents the prior uncertainty in BIOME-BGC parameters. Information about the BIOME-BGC parameters is given in Table 1. SD is effective soil depth

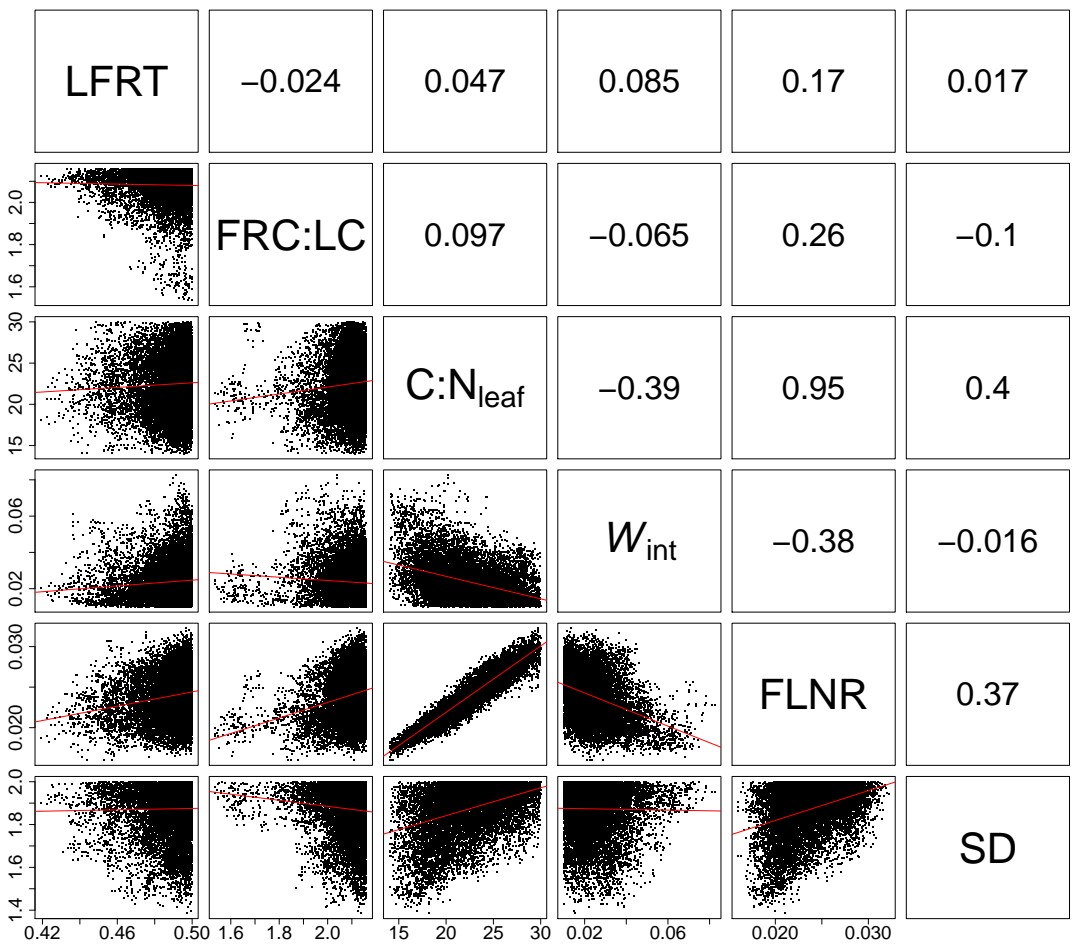

**Figure 3.** Correlation coefficient and scatterplot between the posterior distributions of each pair of calibrated BIOME-BGC parameters obtained from Experiment 1. Information about the BIOME-BGC parameters is given in Table 1. SD is effective soil depth.

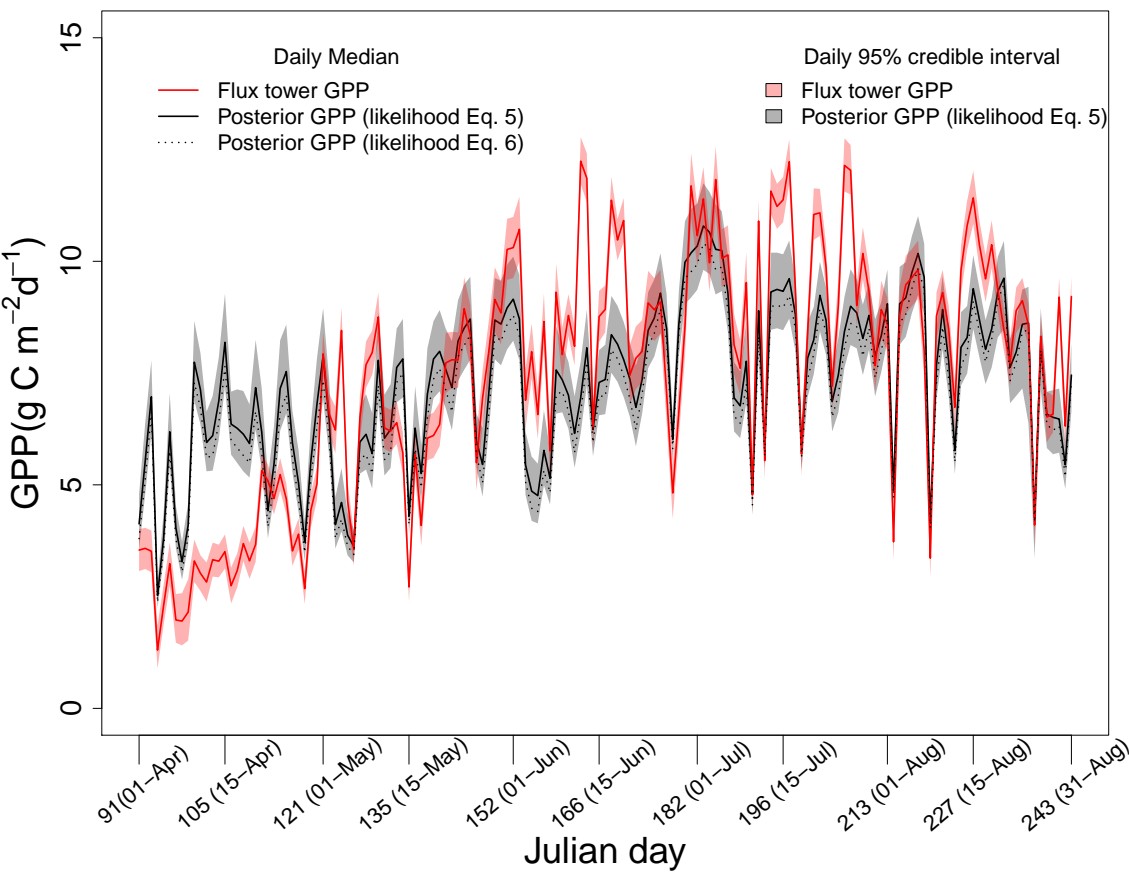

**Figure 4.** Temporal profile of daily posterior GPP, obtained from Experiment 1, and daily flux tower GPP for the calibration period of five months (April to August, Julian days 91 to 243). Daily medians and 95% credible intervals of posterior GPP, obtained using likelihood Eq. 5, are represented by the solid black line and grey shade respectively. Daily medians of posterior GPP, obtained using likelihood Eq. 6, are represented by the dotted black line. Daily medians and 95% credible intervals of flux tower GPP are represented by the red line and light red shade respectively.

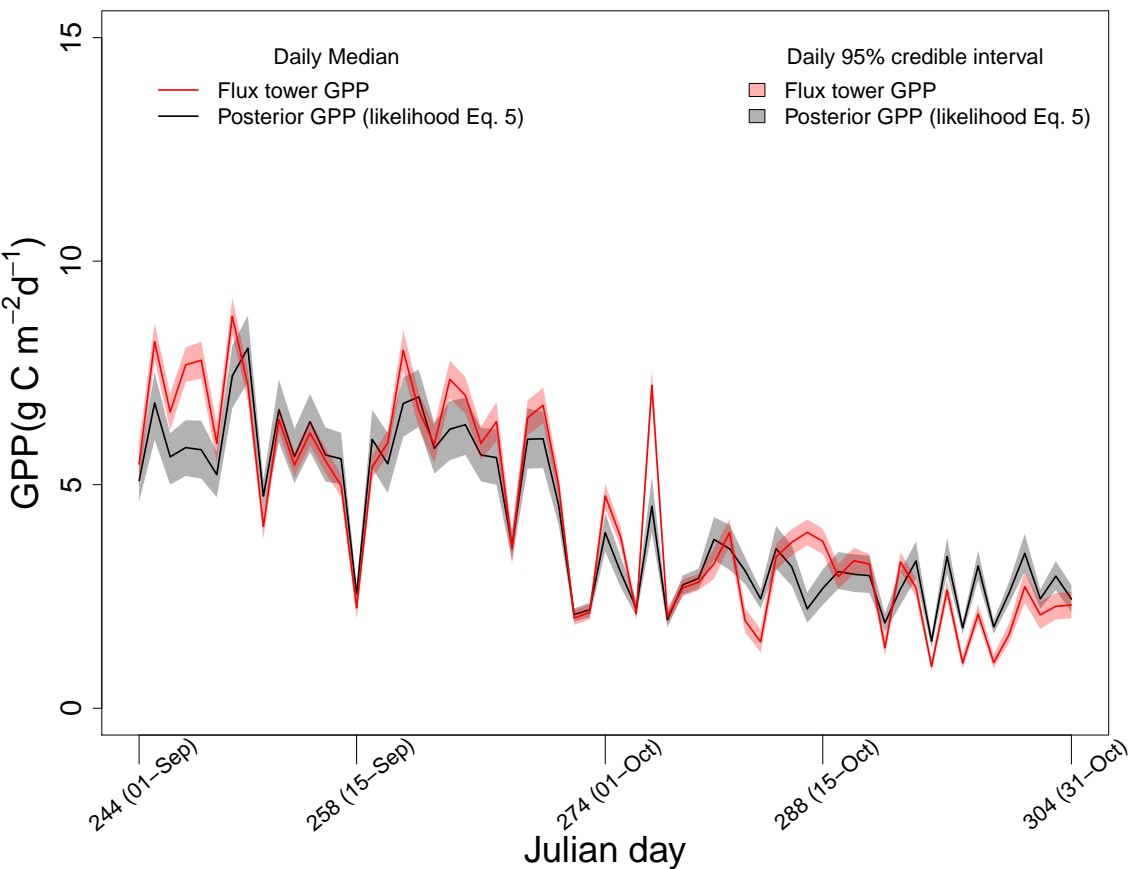

**Figure 5.** Temporal profile of daily posterior GPP , obtained from Experiment 1, and daily flux tower GPP for the validation period of two months (September and October, Julian days 244 to 304). Other details as for Fig. 4.

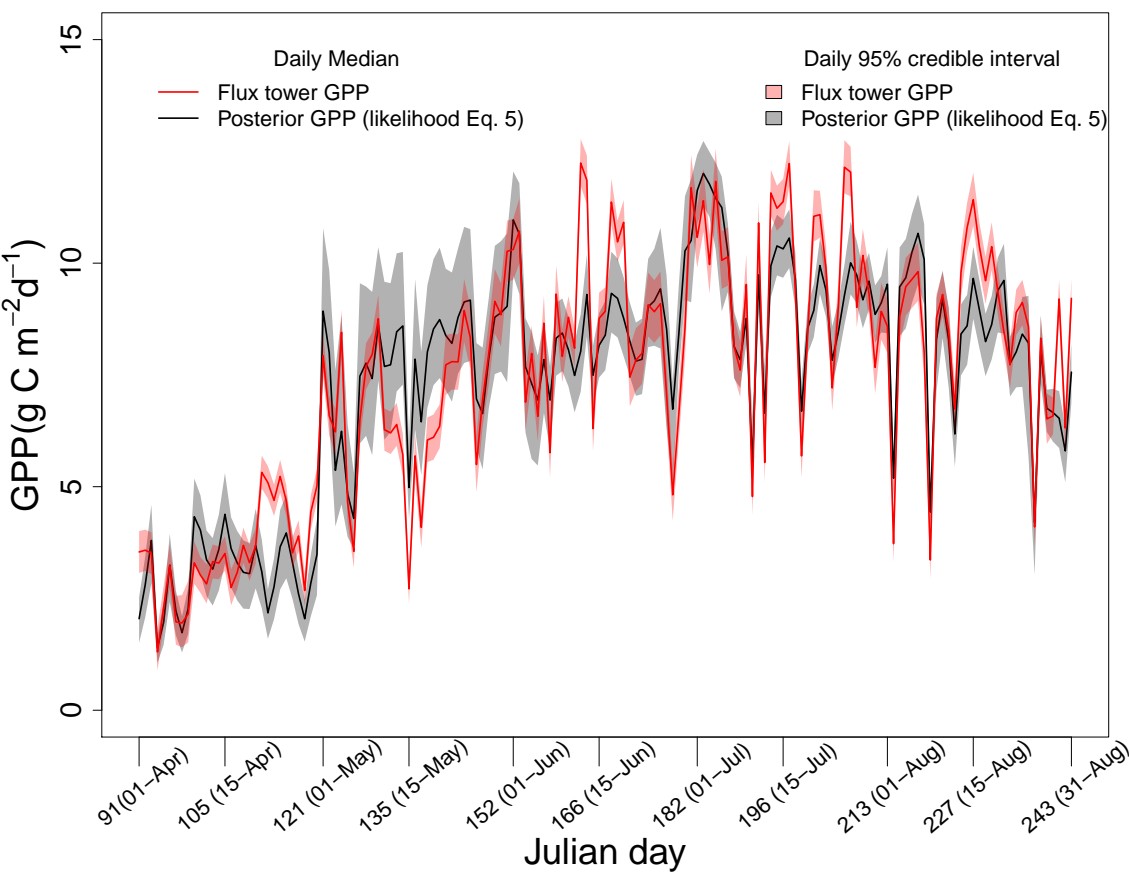

**Figure 6.** Temporal profile of daily posterior GPP, obtained from Experiment 2, and daily flux tower GPP for five months (April to August, Julian days 91 to 243). Other details as for Fig. 4.

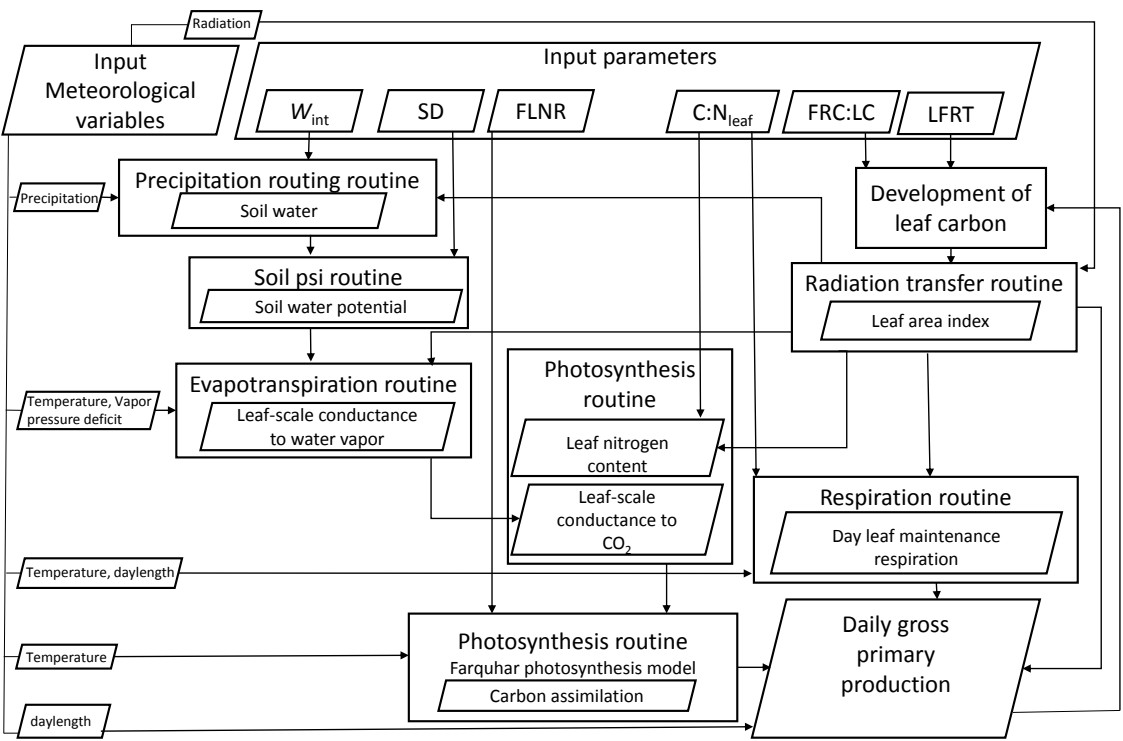

**Figure 7.** The BIOME-BGC internal routines that simulate gross primary production (GPP), controlled by the meteorological data and the six calibrated parameters. Rectangular boxes represent the BIOME-BGC routines and the parallelograms represent the input and output of the routine. Information about the BIOME-BGC parameters is given in Table 1.

**List of Tables**

**Table 1.** 35 ecophysiological parameters needed to run BIOME-BGC for Douglas fir (evergreen needleleaf species). Mean values/distributions were taken from Raj et al. (2014). The ecophysiological parameters highlighted in bold and the soil rooting depth were included in a Bayesian calibration. $U$(min, max), $\mathcal{N}$(mean, standard deviation), $B$(shape1, shape2) represent uniform, normal, and beta distribution respectively.

| Ecophysiological parameter | Symbol | Unit | Mean value/distribution[†] |
|---|---|---|---|
| **Leaf and fine root turnover** | **LFRT** | $1 \text{ yr}^{-1}$ | $U(0.196, 0.5)^{†}$ |
| Annual live wood turnover fraction | LWT | $1 \text{ yr}^{-1}$ | 0.70 |
| Annual whole-plant mortality fraction | WPM | $1 \text{ yr}^{-1}$ | 0.005 |
| Annual fire mortality fraction | FM | $1 \text{ yr}^{-1}$ | 0.005 |
| **new fine root C : new leaf C** | **FRC:LC** | $\text{kg C (kg C)}^{-1}$ | $U(0.78, 2.16)^{†}$ |
| new stem C : new leaf C | SC:LC | $\text{kg C (kg C)}^{-1}$ | 2.391 |
| new live wood C : new total wood C | LWC:TWC | $\text{kg C (kg C)}^{-1}$ | 0.071 |
| new croot C : new stem C | CRC:SC | $\text{kg C (kg C)}^{-1}$ | 0.262 |
| Current growth proportion | CGP | Prop. | 0.5 |
| **C:N of leaves** | **C:N$_{\text{leaf}}$** | $\text{kg C (kg N)}^{-1}$ | $\mathcal{N}(26.731, 3.731)^{†}$ |
| C:N of leaf litter, after retranslocation | C:N$_{\text{lit}}$ | $\text{kg C (kg N)}^{-1}$ | 31.625 |
| C:N of fine roots | C:N$_{\text{fr}}$ | $\text{kg C (kg N)}^{-1}$ | 54.8 |
| C:N of live wood | C:N$_{\text{lw}}$ | $\text{kg C (kg N)}^{-1}$ | 54.8 |
| C:N of dead wood | C:N$_{\text{dw}}$ | $\text{kg C (kg N)}^{-1}$ | 1029.5 |
| Leaf litter labile proportion | L$_{\text{lab}}$ | Unitless | 0.644 |
| Leaf litter cellulose proportion | L$_{\text{cel}}$ | Unitless | 0.201 |
| Leaf litter lignin proportion | L$_{\text{lig}}$ | Unitless | 0.155 |
| Fine root labile proportion | FR$_{\text{lab}}$ | Unitless | 0.527 |
| Fine root cellulose proportion | FR$_{\text{cel}}$ | Unitless | 0.378 |
| Fine root lignin proportion | FR$_{\text{lig}}$ | Unitless | 0.095 |
| Dead wood cellulose proportion | DW$_{\text{cel}}$ | Unitless | 0.772 |
| Dead wood lignin proportion | DW$_{\text{lig}}$ | Unitless | 0.228 |
| **Canopy water interception coefficient** | **W$_{\text{int}}$** | $1 \text{ LAI}^{-1} \text{ day}^{-1}$ | $\mathcal{N}(0.04, 0.02)^{†}$ |
| Canopy light extinction coefficient | k | Unitless | 0.453 |
| All-sided to projected leaf area ratio | LAI$_{\text{all:proj}}$ | $\text{LAI LAI}^{-1}$ | 2.572 |
| Canopy average specific leaf area | SLA | $\text{m}^2 \text{ (kg C)}^{-1}$ | 14.65 |
| Ratio of shaded SLA:sunlit SLA | SLA$_{\text{shd:sun}}$ | $\text{SLA SLA}^{-1}$ | 2.0 |
| **Fraction of leaf N in Rubisco** | **FLNR** | Unitless | $B(25.67, 756.28)^{†}$ |
| Maximum stomatal conductance | $g_{\text{smax}}$ | $\text{m s}^{-1}$ | 0.0051 |
| Cuticular conductance | $g_{\text{cut}}$ | $\text{m s}^{-1}$ | 0.000051 |
| Boundary layer conductance | $g_{\text{bl}}$ | $\text{m s}^{-1}$ | 0.075 |
| Leaf water potential: start of conductance reduction | LWP$_{\text{i}}$ | Mpa | -0.647 |
| Leaf water potential: complete conductance reduction | LWP$_{\text{f}}$ | Mpa | -2.487 |
| Vapor pressure deficit: start of conductance reduction | VPD$_{\text{i}}$ | Pa | 610.0 |
| Vapor pressure deficit: complete conductance reduction | VPD$_{\text{f}}$ | Pa | 3130.0 |
| Site characteristic | | | |
| **Effective soil Depth** | **SD** | meter | $U(0.4, 2)^{†}$ |

**Table 2.** Gelman–Rubin potential scale reduction factor (PSRF) of each BIOME-BGC parameter selected for calibration and $\phi$ (nuisance parameter of likelihood function Eq. 5) for experiment 1 and 2. Information about the BIOME-BGC parameters is given in Table 1. SD is effective soil depth.

| | PSRF | | | | | | | |
| | Experiment 1 | Experiment 2 | | | | | | |
| Parameter ↓ \ Julian days → | 91-243 | 91-120 | 121-151 | 152-181 | 182-212 | 213-243 | 244-273 | 274-304 |
|---|---|---|---|---|---|---|---|---|
| LFRT | 1.05 | 1.03 | 1.01 | 1.03 | 1.03 | 1.04 | 1.03 | 1.01 |
| FRC:LC | 1.09 | 1.02 | 1.01 | 1.01 | 1.04 | 1.06 | 1.02 | 1.01 |
| C:N$_{leaf}$ | 1.04 | 1.02 | 1.02 | 1.01 | 1.04 | 1.04 | 1.03 | 1.03 |
| $W_{\mathbf{int}}$ | 1.02 | 1.01 | 1.01 | 1.02 | 1.06 | 1.03 | 1.03 | 1.01 |
| FLNR | 1.04 | 1.04 | 1.02 | 1.01 | 1.05 | 1.06 | 1.02 | 1.01 |
| SD | 1.04 | 1.03 | 1.02 | 1.02 | 1.03 | 1.1 | 1.02 | 1.02 |
| $\phi$ | 1.02 | 1.03 | 1.01 | 1.01 | 1.03 | 1.03 | 1.01 | 1.01 |

**Table 3.** Root mean square error (RMSE) and Nash-Sutcliffe efficiency (NSE) between the prior (before calibration)/posterior GPP and flux tower GPP for different experiments (see Section 3.3.4) and likelihoods (see Section 3.3.2).

| | | 2.5% | | Median | | 97.5% | |
|---|---|---|---|---|---|---|---|
| | Period | RMSE | NSE | RMSE | NSE | RMSE | NSE |
| Before calibration | April to August | 5.06 | -2.53 | 3.74 | -0.85 | 4.26 | -1.3 |
| | September-October | 2.23 | -0.15 | 1.22 | 0.68 | 2.64 | -0.42 |
| Experiment 1 (With likelihood Eq. 5) | April to August | 1.84 | 0.53 | 1.81 | 0.57 | 1.85 | 0.57 |
| | September-October | 0.91 | 0.81 | 0.83 | 0.85 | 0.79 | 0.87 |
| Experiment 1 (With likelihood Eq. 6) | April to August | 1.82 | 0.54 | 1.87 | 0.54 | 1.94 | 0.52 |
| Experiment 2 (With likelihood Eq. 5) | April to August | 1.3 | 0.77 | 1.24 | 0.8 | 1.45 | 0.73 |
| | September-October | 0.94 | 0.79 | 0.84 | 0.85 | 0.92 | 0.83 |