# Peer review of "Bayesian integration of flux tower data into process-based simulator for quantifying uncertainty in simulated output"

_Geoscientific Model Development, 2016_

## Referee Comment (RC1) · T. Wutzler (Referee) · 26 Oct 2016

The study of Raj et al. presents a successful Bayesian calibration of the biogeochemical model Biome-BGC to Flux-Tower derived Gross Primary Production (GPP). The success of the calibration is shown by several diagnostics and trace plots and by a validation to independent data. Although such Bayesian calibrations of similar models against flux data have been performed before, the aspects of usage of correlated residuals in the cost function and time-varying parameters as well as GPP instead of net ecosystem exchange (NEE) can help for further research. The paper reads well and all the information is given. In order to follow the conclusions some parts are missing, as explained below.

In summary

To my opinion this study presents several aspects that can add to the insight already present by previous studies. But for all of these aspects some more work is required to draw valid conclusions.

**1   Major concens**

**1.1   On conclusion that temporal correlation matters**

A control case without the correlation is missing. How do the results and implications change between accounting versus not accounting for correlations?

**1.2   On conclusion about time varying parameters**

I do not agree with the applied approach. In the presented study several independently simulated time series are mixed together. Each series includes the impact of changed parameters on the previous state. The parameter set valid for July was applied already to April, May, and June and affected the starting states of July. In my opinion one cannot conclude on time-varying parameters with this approach. The simulator needs to be run for the previous months also with the previous parameter set. The model state of the end of the month must be the starting state for the run of the next month with changed parameters. In an ideal case the entire time series would be run as one forward model and the combined (larger) parameter set would be estimated. A more feasible approach is to calibrate each month separately. For the next month calibration continues from a state of the previous month. This starting state needs to be drawn from the distribution of state vectors from the previous month posterior of states for each run with a new parameter sample. For the currently used method, at

minimum, the forward runs that produce the predictive posterior and the fit statistics need to change the parameters across months in each single forward run to discuss seasonally changing parameters.

**1.3 On using GPP to calibrate the mechanistic model**

Net-ecosystem (NEE)-Flux-partitioned GPP is already the output of another statistical model – here the nonrectangular light response curve. This model already makes some strong process assumption e.g. on relationship of respiration with temperature. In effect the mechanistic model is calibrated against the output of another model. This makes it difficult to interpret the estimated parameters, their distribution and their meaning and process understanding. This needs to be discussed.

Biome-BGC also computes respiration and NEE. You can compare these predictions to observations to gain additional insight into the model and the calibration. The flux partitioning also provides seasonally changing respiration at reference temperature and temperature sensitivity. Comparing these quantities to BIOME-BGC predictions lends further insight, which however, may also reveal sub-optimal calibration.

A more direct way would be to include the respiration parts of the Biome-BGC model in the simulation and fit the simulated, i.e. predicted NEE to the NEE observations. Probably, this will introduce correlations in the joint posterior parameter estimates. But the weaker correlations in the presented GPP fit, are only resolved by the assumptions of the NEE-partitioning model that was used to derive GPP.

While the presented GPP calibration has its own ground, those aspect needs to be addressed. The study would greatly benefit from a comparison to a calibration that uses NEE instead of GPP.

**1.4   On hitting the prior bound of residual uncertainty**

Fig 1f clearly shows that the calibration tries to increase the residual variance and that high residual variances yields lower cost. In the current inversion, the residual variance is only bounded by the prior. This hints to deficiencies in the inversion. I sometimes experienced the same effect because a single equation of the cost (eq. 5) may in some cases not prefer the best fitting variance but the larger variance together with sub-optimal parameters. Prescribing an upper bound is to my opinion not a good solution for this problem. Even fixing the residual variance would be a better option. My recommendation is to use several parameter blocks in a Metropolis within Gibbs sampling (Chib S  Greenberg E (1995) Understanding the Metropolis-Hastings algorithm): One block to fit the model parameter conditional on the parameters of the residual statistical distribution and another block to fit the residual distribution parameters conditional on the current sample of model parameters.

**2   Further Concerns**

- The cut of the posterior by the edge of the prior distribution of LFRT and FRC:LC (Fig 1) shows inconsistency in the combination of the model, the prior knowledge, and the observations. This hints to deficiencies of the calibration. It also makes it difficult to interpret the parameter estimates and process understanding. This needs more discussion. The introduction of bias parameters in model drivers or model predictions could help to resolve the inconsistencies and, moreover, the bias parameters then can be interpreted.

- How were the initial states of the model prescribed?

- Do you have correlations in the posterior parameter distribution, and how to you interpret them?

- Please discuss your finding in the context of other studies that already performed a Bayesian calibration of BGC-models against Flux data. E.g. there is big body of studies using the DALEC model also looking at multiple constraints, model error, and different sources uncertainties.

**3   Technical comments**

Fig 1: Shows a very slow mixing. One chain needs more than 1000 steps to become uncorrelated with its previous state. Before computing the Gelman-Rubin criterion you should thin the chains by a factor so that autocorrelation or spectral density of the chain gets small.

Fig 1: shortly explain phi and SD in the figure caption, e.g. "parameters describing variance and correlations of the distribution of model-data residuals (eq. 5)"

Fig 1: Maybe mention, that only the end of the chains after the burnin are shown.

Fig 3,...: Legends are missing. Please, use a different line type so that model and observations can be distinguished without color. Readers would benefit if you indicate months at the time axis instead of or in addition to Julian day.

Fig 6 and associated discussion: For a model with state variables or pools this result is trivial. I suggest omitting or explicitly elaborating on the magnitude of the impacts of state versus drivers on the model output and with witch conditions the one or the other becomes important.

P9L25ff: More discussion needed on hitting the upper prior boundaries and its consequences.

P10L18: typo percentile

P13L25ff: I cannot agree to the discussion because of the method that actually did not

alter parameters across seasons during a single simulation run.

---

## Referee Comment (RC2) · Anonymous Referee #2 · 11 Nov 2016

The manuscript "Bayesian integration of flux tower data into process-based simulator for quantifying uncertainty in simulated output" by R. Raj et al. presents a calibration experiment of six process parameters of the BIOME-BGC terrestrial ecosystem model against GPP data derived from eddy-covariance flux tower measurements. The presented diagnostics (RMSE and NSE) show that the simulation of GPP using the posterior parameter set has improved compared to the prior values. The concept of Bayesian parameter calibration in ecosystem models is not new and has already been demonstrated in many other studies also using eddy-covariance flux tower measurements, but so far has not been applied to BIOME-BGC.

Although the manuscript is mostly well written and rather concise in the presentation of

the methodology it cannot be published in its current form. There are several problems.

The main problem of this manuscript is the use of time-varying parameters. The authors themselves recognise this as a problem (see page 13, lines 11/12). If I understand their use of time-varying parameters correctly ('engineering' a times series of GPP based on independent monthly sub-time series) it actually violates Bayes theorem, mass conservation and model dynamics. Of course one can do such an experiment to better understand the model dynamics and identify missing or mis-represented processes, but the authors are not taking this step and analysing the consequences of their results with the time-varying parameters in terms of model structure and formulation.

Another concern is the use of GPP derived from eddy-covariance flux measurements as the observations in the calibration process. Eddy-covariance towers measure the net exchange flux, essentially NEE, and GPP is the derived from this net flux by employing a model. So essentially, the authors calibrate the BIOME-BGC parameters against another model, in this case the NRH model which makes its own assumptions about the dependency of GPP on environmental conditions.

The whole Section 4.4 is not needed and does not provide any new insights, it is obvious that a dynamical model with state variables such as BIOME-BGC then also depends on its state variables.

So essentially the remaining part of the manuscript concerns experiment 1 and becomes rather light-weighted as a thorough analysis of the results from experiment 1 is missing. For example, how does the posterior error covariance matrix look like and what consequences does this have on the parameters (identifiability) and model? How does the posterior uncertainty compare to prior uncertainty? What is the impact of the observations on other simulated quantities (NEE, NPP), both in terms of their mean and uncertainty? How does the variability and the temporal autocorrelation compare to the prior?

Also the terminology used in the manuscript is somewhat confusing. Sometimes the authors refer to simulated, sometimes to predicted GPP and sometimes to predicted flux tower GPP. In that context they also use the phrase 'posterior flux tower GPP', it is not clear to what the posterior refers?

---

## Author Comment (AC1) · 3 Mar 2017

We thank for the constructive and helpful comments for our manuscript. We will carefully consider all comments and these will be incorporated in our revised manuscript accordingly. We have inserted our response to each comment. We use "R1C" for referee #1's comment and "A1C" for author's response to referee #1.

Major concerns of Referee #1

R1C 1: On conclusion that temporal correlation matters A control case without the correlation is missing. How do the results and implications change between accounting versus not accounting for correlations?

A1C 1: We ran the whole procedure without the temporal correlation in the residuals for Experiment 1. We found that both cases (accounting versus not accounting for correlations) led to similar temporal profile of the posterior simulated gross primary production (GPP) and similar values of statistical criteria (Root mean square error and Nash-Sutcliffe efficiency). The fact that the temporal correlation in the residuals is not only responsible for the temporal development of GPP indicated that the representation of dynamic processes within the BIOME-BGC simulator could be improved. We will add and discuss the new results of the control case without the correlation in the revised manuscript.

R1C 2: On conclusion about time varying parameters I do not agree with the applied approach. In the presented study several independently simulated time series are mixed together. Each series includes the impact of changed parameters on the previous state. The parameter set valid for July was applied already to April, May, and June and affected the starting states of July. In my opinion one cannot conclude on time-varying parameters with this approach. The simulator needs to be run for the previous months also with the previous parameter set. The model state of the end of the month must be the starting state for the run of the next month with changed parameters. In an ideal case the entire time series would be run as one forward model and the combined (larger) parameter set would be estimated. A more feasible approach is to calibrate each month separately. For the next month calibration continues from a state of the previous month. This starting state needs to be drawn from the distribution of state vectors from the previous month posterior of states for each run with a new parameter sample. For the currently used method, at minimum, the forward runs that produce the predictive posterior and the fit statistics need to change the parameters across months in each single forward run to discuss seasonally changing parameters.

A1C 2: We are aware of the problems the reviewer identifies. We found that the response of simulated GPP to weather conditions is rather similar among months: The simulated GPP was mainly driven by the meteorology, and much less by seasonal

phenology. We then hypothesized that some important state variables (such as LAI and carboxylation capacity) may not have a pronounced seasonal cycle in the model. Because these state variables are updated internally in the model, it is not possible to perform a calibration per month: This would require changing the model code, and more importantly, it is in conflict with the main idea of the process simulator. In the second experiment we therefore calibrated the model to the data of each month separately, as if we had no information on GPP for the other months. If some of the parameters have different optimum values when calibrated to different months of data, then this is an indication that the relation between these parameters and important state variables that (should) change during the season, may require improvement.

By doing this experiment, we were able to identify the process that may require an improved description. We will mention above points in the revised manuscript. We decided to avoid the term 'time varying parameters' in the revision.

R1C 3: On using GPP to calibrate the mechanistic model Net-ecosystem (NEE)-Flux-partitioned GPP is already the output of another statistical model – here the nonrectangular light response curve. This model already makes some strong process assumption e.g. on relationship of respiration with temperature. In effect the mechanistic model is calibrated against the output of another model. This makes it difficult to interpret the estimated parameters, their distribution and their meaning and process understanding. This needs to be discussed. Biome-BGC also computes respiration and NEE. You can compare these predictions to observations to gain additional insight into the model and the calibration. The flux partitioning also provides seasonally changing respiration at reference temperature and temperature sensitivity. Comparing these quantities to BIOME-BGC predictions lends further insight, which however, may also reveal sub-optimal calibration. A more direct way would be to include the respiration parts of the Biome-BGC model in the simulation and fit the simulated, i.e. predicted NEE to the NEE observations. Probably, this will introduce correlations in the joint posterior parameter estimates. But the weaker correlations in the presented

GPP fit, are only resolved by the assumptions of the NEE-partitioning model that was used to derive GPP. While the presented GPP calibration has its own ground, those aspect needs to be addressed. The study would greatly benefit from a comparison to a calibration that uses NEE instead of GPP.

A1C 3: We would like to mention following points on using partitioned GPP instead of NEE data in this study:

A. Indeed in our approach, the output of the process-based simulator was validated against the output of another model, notably the flux partitioning model. Although this approach has been used in other studies to validate the output of process-based simulator as well (Collalti et al., 2016, Liu et al., 2014, Yuan et al., 2014, Zhou et al., 2016), it could lead to error propagation. We clarify that the flux partitioning model (NRH model, Raj et al., (2016)) was tuned to the Eddy Covariance data in blocks of 10 days. Because the NHR and the relationship of respiration with temperature and moisture were tuned for these short blocks separately, we expect that the GPP still reflects realistic responses to environmental drivers, and does not depend much on model assumptions.

B. Calibration of BIOME-BGC using NEE data is more challenging as NEE is the difference between fluxes caused by two processes (assimilation and respiration) We argue that calibration of such a complex model to NEE instead of GPP may not be a good idea, but calibration to NEE or respiration in addition to GPP is possible. However, we limited this study to the primary productivity, because this was our primary interest. A future study should be done to include both GPP data and ecosystem respiration data (can also be achieved by partitioning of NEE data) in a Bayesian calibration of BIOME-BGC. This may ensure the accuracy of all related carbon budget terms (GPP, NEE, and respiration terms).

C. As far as the comparison of simulated NEE and respiration with the measured NEE and portioned respiration is concerned, this is out of the scope of the present study.

We mainly focused on the simulated GPP and already presented a lot of results.

We will discuss all above points in our revised manuscript.

R1C 4: On hitting the prior bound of residual uncertainty Fig 1f clearly shows that the calibration tries to increase the residual variance and that high residual variances yields lower cost. In the current inversion, the residual variance is only bounded by the prior. This hints to deficiencies in the inversion. I sometimes experienced the same effect because a single equation of the cost (eq. 5) may in some cases not prefer the best fitting variance but the larger variance together with suboptimal parameters. Prescribing an upper bound is to my opinion not a good solution for this problem. Even fixing the residual variance would be a better option. My recommendation is to use several parameter blocks in a Metropolis within Gibbs sampling (Chib S Greenberg E (1995) Understanding the Metropolis-Hastings algorithm): One block to fit the model parameter conditional on the parameters of the residual statistical distribution and another block to fit the residual distribution parameters conditional on the current sample of model parameters.

A1C 4: We would like to clarify that SD in Fig 1 of the manuscript is not the residual variance. SD represents the effective soil depth and this is considered in this study as one of the parameters of BIOME-BGC simulator. In the figure cation, we have written that "Information about the BIOME-BGC parameters is given in Table 1". We will specifically mention in the figure caption that "SD is effective soil depth" in the revised manuscript to avoid any confusion.

Further Concerns of Referee #1

R1C 5: The cut of the posterior by the edge of the prior distribution of LFRT and FRC:LC (Fig 1) shows inconsistency in the combination of the model, the prior knowledge, and the observations. This hints to deficiencies of the calibration. It also makes it difficult to interpret the parameter estimates and process understanding. This needs more discussion. The introduction of bias parameters in model drivers or model predictions could help to resolve the inconsistencies and, moreover, the bias parameters then can be interpreted.

A1C 5: Fig 1 shows the cut of the posterior by the edge of the prior distribution, which is called edge effect, of LFRT and FRC:LC. We agree that this clearly indicates a significant effect on our posterior parameter space of LFRT and FRC:LC by our particular choices of parameter space to be included in the prior distributions of LFRT and FRC:LC. It could be argued that the prior uniform distributions of LFRT and FRC:LC could be made wider in order to eliminate the edge effect. However, we did not do this for the following reasons:

A. We carried out extensive literature review in a previous study (Raj et al., 2014) to compile the information on FRC:LC, LFRT, and other BIOME-BGC parameters. This information led to the characterization of uncertainty in the parameters and helped to define the prior distributions. For the present study, we used this information on prior distributions. We had no further scope to make the prior distributions wider at the study site.

B. In the present study, we have used the upper limit of FRC:LC at 2.15. In our previous study (Raj et al., 2014) we found that higher values of FRC:LC led to disappearance of the forest (LAI=0) due to negative cumulative NEE, and hence no production at the study site. Therefore, we had no other choice of the upper limit of FRC:LC other than 2.16.

Further, we don't fully agree that the edge effect indicates deficiencies of the calibration. This can also be thought in another way that even if there is the edge effect, a drop in RMSE and enhancement in NSE coefficient (Table 3 in the manuscript) before and after calibration indicated the efficiency of the calibration. We will mention above points in the revised manuscript.

R1C 6: How were the initial states of the model prescribed?

A1C 6: Initial states of the model were prescribed with very low value ($\approx$ 0). Spin up simulation of BIOME-BGC was performed first to achieve steady state condition of soil carbon and nitrogen pools under given climate and site condition. Normal simulation was then started with these steady state condition using daily meteorological data of 2009. We will add these points in the revised manuscript.

R1C 7: Do you have correlations in the posterior parameter distribution, and how to you interpret them?

A1C 7: In the revised manuscript, we will add a brief explanation and a plot showing the correlations in the posterior parameter distribution obtained in Experiment 1. We found a strong positive correlation between the posterior distributions of C:Nleaf (carbon and nitrogen ratio) and FLNR (Fraction of leaf N in Rubisco) with r=0.95 (r is correlation coefficient). This strong positive correlation is in-line with the formulation of FLNR that shows direct proportionality with C:Nleaf (see Appendix A in RajEtal2014, for details). The parameters C:Nleaf and FLNR showed similar negative, but weak (> -0.5), correlation with Wint (Canopy water interception coefficient) (r $\approx$ -0.4). This can be explained by the fact that the simulated GPP is expected to vary inversely with Wint via soil water potential and stomatal regulation and directly with FLNR and C:Nleaf (see Section 5.1 in the manuscript, for details of BIOME-BGC internal routines). The parameter SD (effective soil depth) had similar positive, but weak (< 0.5), correlation with FLNR and C:Nleaf (r $\approx$ 0.4). This can be explained by the fact that the simulated GPP is expected to vary directly with SD (via soil water potential and stomatal regulation), and FLNR and C:Nleaf .Two parameters of any other pair combinations did not show any notable correlation.

R1C 8: Please discuss your finding in the context of other studies that already performed a Bayesian calibration of BGC-models against Flux data. E.g. there is big body of studies using the DALEC model also looking at multiple constraints, model error, and different sources uncertainties.

[Figure]

A1C 8: We couldn't find papers on a Bayesian calibration of BIOME-BGC against flux data that compare with our results directly. We found other papers on calibration of BIOME-BGC, and one study (Hidy et al., 2012) on Bayesian calibration of BIOME-BGC. Because Hidy et al. (2012) focussed on an herbaceous ecosystem, we could not compare directly the outcome quantitatively.

Technical comments of Referee #1

R1C 9: Fig 1: Shows a very slow mixing. One chain needs more than 1000 steps to become uncorrelated with its previous state. Before computing the Gelman-Rubin criterion you should thin the chains by a factor so that autocorrelation or spectral density of the chain gets small.

A1C 9: We agree with the referee that thinning would reduce the spectral density of the chains. We decided not to do it again for the revised manuscript because we expect that thinning will not change the posterior estimations achieved in the study.

R1C 10: Fig 1: shortly explain phi and SD in the figure caption, e.g. "parameters describing variance and correlations of the distribution of model-data residuals (eq. 5)"

A1C 10: Please refer to our comment A1C 4 on the clarification of SD. We will explain phi in the figure caption in the revised manuscript.

R1C 11: Fig 1: Maybe mention, that only the end of the chains after the burnin are shown.

A1C 11: Thank you very much for this suggestion. We will mention this in the figure caption in the revised manuscript.

R1C 12: Fig 3,. . .: Legends are missing. Please, use a different line type so that model and observations can be distinguished without color. Readers would benefit if you indicate months at the time axis instead of or in addition to Julian day.

A1C 12: We will modify the figure in the revised manuscript according to the referee's
suggestion.

R1C 13: Fig 6 and associated discussion: For a model with state variables or pools this result is trivial. I suggest omitting or explicitly elaborating on the magnitude of the impacts of state versus drivers on the model output and with witch conditions the one or the other becomes important.

A1C 13: We will omit figure 6 in our revised manuscript.

R1C 14: P9L25ff: More discussion needed on hitting the upper prior boundaries and its consequences.

A1C 14: Please refer to our comment A1C 5.

R1C 15: P10L18: typo percentile

A1C 15: We will correcte this in the revised manuscript.

R1C 16: I cannot agree to the discussion because of the method that actually did not alter parameters across seasons during a single simulation run.

A1C 16: Please refer to our comment A1C 2.

References:

Collalti, A., Marconi, S., Ibrom, A., Trotta, C., Anav, A., D'Andrea, E., Matteucci, G., Montagnani, L., Gielen, B., Mammarella, I., Grünwald, T., Knohl, A., Berninger, F., Zhao, Y., Valentini, R., and Santini, M.: Validation of 3D-CMCC forest ecosystem model (v.5.1) against eddy covariance data for 10 European forest sites, Geoscientific Model Development, 9, 479–504, 2016.

Hidy, D., Barcza, Z., Haszpra, L., Churkina, G., Pintér, K., & Nagy, Z.: Development of the Biome-BGC model for simulation of managed herbaceous ecosystems, Ecological Modelling, 226, 99-119, 2012.

Raj, R., Hamm, N. A. S., van der Tol, C., and Stein, A.: Variance-based sensitivity

analysis of BIOME-BGC for gross and net primary production, Ecological Modelling, 292, 26–36, 2014.

Raj, R., Hamm, N. A. S., van der Tol, C., and Stein, A.: Uncertainty analysis of gross primary production partitioned from net ecosystem exchange measurements, Biogeosciences, 13, 1409–1422, doi:10.5194/bg-13-1409-2016, 2016.

Liu, D., Cai, W., Xia, J., Dong, W., Zhou, G., Chen, Y., Zhang, H., and Yuan, W.: Global validation of a process-based model on vegetation gross primary production using eddy covariance observations, PLOS ONE, 9, e110 407, 2014.

Yuan, W., Cai, W., Xia, J., Chen, J., Liu, S., Dong, W., Merbold, L., Law, B., Arain, A., Beringer, J., Bernhofer, C., Black, A., Blanken, P. D., Cescatti, A., Chen, Y., Francois, L., Gianelle, D., Janssens, I. A., Jung, M., Kato, T., Kiely, G., Liu, D., Marcolla, B., Montagnani, L., Raschi, A., Roupsard, O., Varlagin, A., and Wohlfahrt, G.: Global comparison of light use efficiency models for simulating terrestrial vegetation gross primary production based on the LaThuile database, Agricultural and Forest Meteorology, 192—193, 108–120, 2014.

Zhou, Y., Wu, X., Ju, W., Chen, J. M., Wang, S., Wang, H., Yuan, W., Andrew Black, T., Jassal, R., Ibrom, A., Han, S., Yan, J., Margolis, H., Roupsard, O., Li, Y., Zhao, F., Kiely, G., Starr, G., Pavelka, M., Montagnani, L., Wohlfahrt, G., D'Odorico, P., Cook, D., Arain, M. A., Bonal, D., Beringer, J., Blanken, P. D., Loubet, B., Leclerc, M. 5 Y., Matteucci, G., Nagy, Z., Olejnik, J., Paw U, K. T., and Varlagin, A.: Global parameterization and validation of a two-leaf light use efficiency model for predicting gross primary production across FLUXNET sites, Journal of Geophysical Research: Biogeosciences, 121, 1045–1072, 2016.

---

## Author Comment (AC2) · 3 Mar 2017

We thank for the constructive and helpful comments for our manuscript. We will carefully consider all comments and these will be incorporated in our revised manuscript accordingly. We have inserted our response to each comment. We use "R2C" for referee #2's comment and "A2C" for author's response to referee #2.

R2C 1: The main problem of this manuscript is the use of time-varying parameters. The authors themselves recognise this as a problem (see page 13, lines 11/12). If I understand their use of time-varying parameters correctly ('engineering' a times series of GPP based on independent monthly sub-time series) it actually violates Bayes theorem, mass conservation and model dynamics. Of course one can do such an experi-

ment to better understand the model dynamics and identify missing or misrepresented processes, but the authors are not taking this step and analysing the consequences of their results with the time-varying parameters in terms of model structure and formulation.

A2C 1: We understand the concern of referee about the use time varying parameters. We found that the response of simulated GPP to weather conditions is rather similar among months: The simulated GPP was mainly driven by the meteorology, and much less by seasonal phenology. We then hypothesized that some important state variables (such as LAI and carboxylation capacity) may not have a pronounced seasonal cycle in the model. In the second experiment, we calibrated Biome-BGC to the GPP of each month separately, as if the data for the other months did not exist. In that way neither mass conservation nor the Bayes theorem is violated. If some of the parameters have different optimum values when calibrated to different months of data, then this is an indication that the relation between these parameters and important state variables that (should) change during the season, may require improvement. The problem only arises when we combine the results in a time series. We then merge different simulations outputs into one. The objective was indeed to better understand the dynamics (or lack of dynamics) of the model. We decided to avoid the term 'time varying parameters' in the revision. We will mention above points in the revised manuscript.

R2C 2: Another concern is the use of GPP derived from eddy-covariance flux measurements as the observations in the calibration process. Eddy-covariance towers measure the net exchange flux, essentially NEE, and GPP is the derived from this net flux by employing a model. So essentially, the authors calibrate the BIOME-BGC parameters against another model, in this case the NRH model which makes its own assumptions about the dependency of GPP on environmental conditions.

A2C 2: We would like to mention following points on using partitioned GPP instead of NEE data in this study:

A. Indeed in our approach, the output of the process-based simulator was validated against the output of another model, notably the flux partitioning model. Although this approach has been used in other studies to validate the output of process-based simulator as well (Collalti et al., 2016, Liu et al., 2014, Yuan et al., 2014, Zhou et al., 2016), it could lead to error propagation. We clarify that the flux partitioning model (NRH model, Raj et al., (2016)) was tuned to the Eddy Covariance data in blocks of 10 days. Because the NHR and the relationship of respiration with temperature and moisture were tuned for these short blocks separately, we expect that the GPP still reflects realistic responses to environmental drivers, and does not depend much on model assumptions.

B. Calibration of BIOME-BGC using NEE data is more challenging as NEE is the difference between fluxes caused by two processes (assimilation and respiration) We argue that calibration of such a complex model to NEE instead of GPP may not be a good idea, but calibration to NEE or respiration in addition to GPP is possible. However, we limited this study to the primary productivity, because this was our primary interest. A future study should be done to include both GPP data and ecosystem respiration data (can also be achieved by partitioning of NEE data) in a Bayesian calibration of BIOME-BGC. This may ensure the accuracy of all related carbon budget terms (GPP, NEE, and respirations).

We will discuss all above points in our revised manuscript.

R2C 3: The whole Section 4.4 is not needed and does not provide any new insights, it is obvious that a dynamical model with state variables such as BIOME-BGC then also depends on its state variables.

A2C 3: We will remove section 4.4 in the revised manuscript.

R2C 4: So essentially the remaining part of the manuscript concerns experiment 1 and becomes rather light-weighted as a thorough analysis of the results from experiment 1 is missing. For example, how does the posterior error covariance matrix look like and

what consequences does this have on the parameters (identifiability) and model? How does the posterior uncertainty compare to prior uncertainty? What is the impact of the observations on other simulated quantities (NEE, NPP), both in terms of their mean and uncertainty? How does the variability and the temporal autocorrelation compare to the prior?

A2C 4:

A. In the revised manuscript, we will add a brief explanation and a plot showing the correlations in the posterior parameter distributions obtained in Experiment 1. We found a strong positive correlation between the posterior distributions of C:Nleaf (carbon and nitrogen ratio) and FLNR (Fraction of leaf N in Rubisco) with r=0.95 (r is correlation coefficient). This strong positive correlation is in-line with the formulation of FLNR that shows direct proportionality with C:Nleaf (see Appendix A in Raj et al., 2014, for details). The parameters C:Nleaf and FLNR showed similar negative, but weak (> -0.5), correlation with Wint (Canopy water interception coefficient) (r ≈ -0.4). This can be explained by the fact that the simulated GPP is expected to vary inversely with Wint via soil water potential and stomatal regulation and directly with FLNR and C:Nleaf (see Section 5.1 in the manuscript, for details of BIOME-BGC internal routines). The parameter SD (effective soil depth) had similar positive, but weak (< 0.5), correlation with FLNR and C:Nleaf (r ≈ 0.4). This can be explained by the fact that the simulated GPP is expected to vary directly with SD (via soil water potential and stomatal regulation), and FLNR and C:Nleaf .Two parameters of any other pair combinations did not show any notable correlation.

B. We have already compared posterior and prior uncertainty in the section 4.2 of the manuscript (Fig 2 and P9 L22-25).

C. As far as the impact of the observations on other simulated quantities (NEE, NPP) is concerned, this is out of the scope of the present study. We mainly focused on the simulated GPP and already presented a lot of results.

D. This study modelled the temporal correlation in the residuals during the calibration by adding the nuisance parameter ÑĎ in the likelihood function (see Section 3.3.2 in the manuscript). We had assumed uniform prior distribution of phi between -1 and +1. In the posterior, we obtained the range of phi from 0.56 to 0.93 with a mean at 0.75 (Fig 1g and P9 L15-18 in the manuscript). This showed the reduction in posterior uncertainty in phi compared to prior. We will mention the choice of prior uncertainty in phi in the revised manuscript.

R2C 5: Also the terminology used in the manuscript is somewhat confusing. Sometimes the authors refer to simulated, sometimes to predicted GPP and sometimes to predicted flux tower GPP. In that context they also use the phrase 'posterior flux tower GPP', it is not clear to what the posterior refers?

A2C 5: We apologize for this confusion. We would like to clarify that the term "posterior" refers to the GPP obtained with posterior distribution of parameters. We agree that we had not used the terminology consistently. In the revised manuscript, we will make the terminology consistent as mentioned below:

A. "Flux tower GPP" - We will use this single term throughout the revised manuscript to indicate GPP partitioned from flux tower observation of net ecosystem exchange.

B. "Posterior GPP" - We will use this single term throughout the revised manuscript to indicate GPP simulated by BIOME-BGC at the posterior distribution of parameters.

C. "Prior GPP" - We will use this single term throughout the revised manuscript to indicate GPP simulated by BIOME-BGC at the prior distribution of parameters.

D. "Simulated GPP" – Sometimes, We will use this term in the revised manuscript to give the general description of GPP simulated by BIOME-BGC irrespective of GPP simulated at prior or posterior distributions.

We will clarify this in the revised manuscript.

References:

Collalti, A., Marconi, S., Ibrom, A., Trotta, C., Anav, A., D'Andrea, E., Matteucci, G., Montagnani, L., Gielen, B., Mammarella, I., Grünwald, T., Knohl, A., Berninger, F., Zhao, Y., Valentini, R., and Santini, M.: Validation of 3D-CMCC forest ecosystem model (v.5.1) against eddy covariance data for 10 European forest sites, Geoscientific Model Development, 9, 479–504, 2016.

Raj, R., Hamm, N. A. S., van der Tol, C., and Stein, A.: Variance-based sensitivity analysis of BIOME-BGC for gross and net primary production, Ecological Modelling, 292, 26–36, 2014.

Raj, R., Hamm, N. A. S., van der Tol, C., and Stein, A.: Uncertainty analysis of gross primary production partitioned from net ecosystem exchange measurements, Biogeosciences, 13, 1409–1422, doi:10.5194/bg-13-1409-2016, 2016.

Liu, D., Cai, W., Xia, J., Dong, W., Zhou, G., Chen, Y., Zhang, H., and Yuan, W.: Global validation of a process-based model on vegetation gross primary production using eddy covariance observations, PLOS ONE, 9, e110 407, 2014.

Yuan, W., Cai, W., Xia, J., Chen, J., Liu, S., Dong, W., Merbold, L., Law, B., Arain, A., Beringer, J., Bernhofer, C., Black, A., Blanken, P. D., Cescatti, A., Chen, Y., Francois, L., Gianelle, D., Janssens, I. A., Jung, M., Kato, T., Kiely, G., Liu, D., Marcolla, B., Montagnani, L., Raschi, A., Roupsard, O., Varlagin, A., and Wohlfahrt, G.: Global comparison of light use efficiency models for simulating terrestrial vegetation gross primary production based on the LaThuile database, Agricultural and Forest Meteorology, 192-193, 108–120, 2014.

Zhou, Y., Wu, X., Ju, W., Chen, J. M., Wang, S., Wang, H., Yuan, W., Andrew Black, T., Jassal, R., Ibrom, A., Han, S., Yan, J., Margolis, H., Roupsard, O., Li, Y., Zhao, F., Kiely, G., Starr, G., Pavelka, M., Montagnani, L., Wohlfahrt, G., D'Odorico, P., Cook, D., Arain, M. A., Bonal, D., Beringer, J., Blanken, P. D., Loubet, B., Leclerc, M. 5 Y., Matteucci, G., Nagy, Z., Olejnik, J., Paw U, K. T., and Varlagin, A.: Global parameterization and validation of a two-leaf light use efficiency model for predicting gross primary production across FLUXNET sites, Journal of Geophysical Research: Biogeosciences, 121, 1045–1072, 2016.

---

## Author Response (AR1)

Dr. Rahul Raj

Faculty of Geo-Information science and Earth Observation (ITC), University of Twente,

PO Box 217, 7500AE, Enschede, The Netherlands

Email: r.raj@utwente.nl

Date: 07 March 2017

Dear Philippe Peylin,

We are pleased to submit the revised version of our manuscript titled "Bayesian integration of flux tower data into process-based simulator for quantifying uncertainty in simulated output". We appreciated the constructive criticisms of both referees that have helped to improve the manuscript. The response to the referees is given underneath this letter. We hope that we have addressed the comments to the satisfaction of both referees.

In general we have extended the relevant sections in order to address the referees' comments. We have referred to these changes in the response to the referees. Unless specifically requested by the referees, we have not substantively altered the content or format of the paper. In the revised manuscript, new text is highlighted in red. Deleted text is indicated by strikethrough. The term "previous manuscript" used below refers to published Geoscientific Model Development discussion paper. The term "revised manuscript" refers to the manuscript attached at the end.

We would like to clarify some points:

1.  The most important concerns of both referees are the use of "time varying parameters of BIOME-BGC simulator" and "GPP partitioned from the flux tower measurements of net ecosystem exchange (NEE) in a Bayesian calibration". We fully understand the concerns of both referees. Regarding time varying parameters (obtained in month wise calibration), we agree with referees that this violates the basic assumption of BIOME-BGC. A real month by month calibration as suggested by referee #1 (see the comment ''R1C 2'' below) is not possible in BIOME-BGC precisely because it is a dynamic vegetation simulator. Without modifying BIOME-BGC itself, we could only force BIOME-BGC to simulate more flexible state variables (such as leaf area index (LAI) and carboxylation capacity) in an artificial way (obtaining calibrated parameters for each month). The objective of this exercise was to identify the aspects of the model (i.e. relations between parameters and state variables) that may require improvement in order to better simulate the shape of the seasonal cycle of GPP. We acknowledge that this was not clear in the original manuscript, and we have elaborated this in the revision. Please see our response ''**A1C 2**'' below for more details.

2.  The main idea behind the use of partitioned GPP instead of NEE data (measured on flux tower) directly in a Bayesian calibration was to tune the BIOME-BGC simulator parameters specifically for simulated GPP, which is the main output variable of interest in the paper. Please see our response "**A1C 3**" below for more details.

3. In the revised manuscript, we have deleted the section 4.4 and associated figures 5 and 6 of the previous manuscript following both referees' comments (**R1C 13** and **R2C 3** below). We have also added one extra figure (labelled Figure 3 in the revised manuscript) following referee's comment (**R1C 7** and **R2C 4** below)

4. We have deleted and revised some figures in the manuscript as follows:

| Old figure number in the previous manuscript | New figure number in the revised manuscript | Remark |
|---|---|---|
| Figure 1 | Figure 1 | Both figures in the previous and revised manuscripts are same. |
| Figure 2 | Figure 2 | Both figures in the previous and revised manuscripts are same. |
| Figure 3 | Figure 4 | Figure 3 in the previous manuscript is revised following referee #1 comment **R1C 12** (see below) |
| Figure 4 | Figure 5 | Figure 4 in the previous manuscript is revised following referee #1 comment **R1C 12** (see below) |
| Figure 5 | Removed | |
| Figure 6 | Removed | |
| Figure 7 | Figure 6 | Figure 7 in the previous manuscript is revised following referee #1 comment **R1C 12** (see below) |
| Figure 8 | Figure 7 | Both figures in the previous and revised manuscripts are same. |

5. We have extended table 3 (by adding one extra column for Experiment 1) to address referee #1 comment **R1C 1** (see below).

We look forward to hearing from you.

Yours sincerely

Dr Rahul Raj
(On behalf of all authors)

**Response to Referee #1 for "Bayesian integration of flux tower data into process-based simulator for quantifying uncertainty in simulated output" by R. Raj et al.**

We thank for the constructive and helpful comments for our manuscript. We have carefully considered all comments and these have incorporated in our revised manuscript accordingly. We have inserted our response to each comment. We use "R1C" for referee #1's comment and "A1C" for author's response to referee #1.

Major concerns of Referee #1

**R1C 1:** On conclusion that temporal correlation matters
A control case without the correlation is missing. How do the results and implications change between accounting versus not accounting for correlations?

**A1C 1:** We ran the whole procedure without the temporal correlation in the residuals for Experiment 1. We found that both cases (accounting versus not accounting for correlations) led to similar temporal profile of the posterior simulated gross primary production (GPP) and similar values of statistical criteria (Root mean square error and Nash-Sutcliffe efficiency). The fact that the temporal correlation in the residuals is not only responsible for the temporal development of GPP indicated that the representation of dynamic processes within the BIOME-BGC simulator could be improved. We have added the new results of the control case without the correlation (see Figure 4 in the revised manuscript). We have explained the results in our revised manuscript (P7 L23-29, P8 L22, P12 L1-5, P13 L32-33, P14 L1-2, and P16 L3-8).

**R1C 2:** On conclusion about time varying parameters
I do not agree with the applied approach. In the presented study several independently simulated time series are mixed together. Each series includes the impact of changed parameters on the previous state. The parameter set valid for July was applied already to April, May, and June and affected the starting states of July. In my opinion one cannot conclude on time-varying parameters with this approach. The simulator needs to be run for the previous months also with the previous parameter set. The model state of the end of the month must be the starting state for the run of the next month with changed parameters. In an ideal case the entire time series would be run as one forward model and the combined (larger) parameter set would be estimated. A more feasible approach is to calibrate each month separately. For the next month calibration continues from a state of the previous month. This starting state needs to be drawn from the distribution of state vectors from the previous month posterior of states for each run with a new parameter sample. For the currently used method, at minimum, the forward runs that produce the predictive posterior and the fit statistics need to change the parameters across months in each single forward run to discuss seasonally changing parameters.

**A1C 2:** We are aware of the problems the reviewer identifies. We found that the response of simulated GPP to weather conditions is rather similar among months: The simulated GPP was mainly driven by the meteorology, and much less by seasonal phenology. We then hypothesized that some important state variables (such as LAI and carboxylation capacity) may not have a pronounced seasonal cycle in the model. Because these state variables are updated internally in the model, it is

not possible to perform a calibration per month: This would require changing the model code, and more importantly, it is in conflict with the main idea of the process simulator. In the second experiment we therefore calibrated the model to the data of each month separately, as if we had no information on GPP for the other months. If some of the parameters have different optimum values when calibrated to different months of data, then this is an indication that the relation between these parameters and important state variables that (should) change during the season, may require improvement.

By doing this experiment, we were able to identify the process that may require an improved description. We have mentioned above points in the revised manuscript (P14 L15-24, and P14 L25-35, and P15 L1-9). We decided to avoid the term 'time varying parameters' in the revision.

**R1C 3:** On using GPP to calibrate the mechanistic model
Net-ecosystem (NEE)-Flux-partitioned GPP is already the output of another statistical model – here the nonrectangular light response curve. This model already makes some strong process assumption e.g. on relationship of respiration with temperature. In effect the mechanistic model is calibrated against the output of another model. This makes it difficult to interpret the estimated parameters, their distribution and their meaning and process understanding. This needs to be discussed. Biome-BGC also computes respiration and NEE. You can compare these predictions to observations to gain additional insight into the model and the calibration. The flux partitioning also provides seasonally changing respiration at reference temperature and temperature sensitivity. Comparing these quantities to BIOME-BGC predictions lends further insight, which however, may also reveal sub-optimal calibration. A more direct way would be to include the respiration parts of the Biome-BGC model in the simulation and fit the simulated, i.e. predicted NEE to the NEE observations. Probably, this will introduce correlations in the joint posterior parameter estimates. But the weaker correlations in the presented GPP fit, are only resolved by the assumptions of the NEE-partitioning model that was used to derive GPP. While the presented GPP calibration has its own ground, those aspect needs to be addressed. The study would greatly benefit from a comparison to a calibration that uses NEE instead of GPP.

**A1C 3:** We would like to mention following points on using partitioned GPP instead of NEE data in this study:

A. Indeed in our approach, the output of the process-based simulator was validated against the output of another model, notably the flux partitioning model. Although this approach has been used in other studies to validate the output of process-based simulator as well (Collalti et al., 2016, Liu et al., 2014, Yuan et al., 2014, Zhou et al., 2016), it could lead to error propagation. We clarify that the flux partitioning model (NRH model, Raj et al., (2016)) was tuned to the Eddy Covariance data in blocks of 10 days. Because the NHR and the relationship of respiration with temperature and moisture were tuned for these short blocks separately, we expect that the GPP still reflects realistic responses to environmental drivers, and does not depend much on model assumptions.

B. Calibration of BIOME-BGC using NEE data is more challenging as NEE is the difference between fluxes caused by two processes (assimilation and respiration) We argue that calibration of such a complex model to NEE *instead* of GPP may not be a good idea, but calibration to NEE or respiration in addition to GPP is possible. However, we limited this study to the primary productivity, because this was our primary interest. A future study should be done to include both GPP data and ecosystem respiration data (can also be

achieved by partitioning of NEE data) in a Bayesian calibration of BIOME-BGC. This may ensure the accuracy of all related carbon budget terms (GPP, NEE, and respiration terms).

C. As far as the comparison of simulated NEE and respiration with the measured NEE and portioned respiration is concerned, this is out of the scope of the present study. We mainly focused on the simulated GPP and already presented a lot of results.

We have discussed all above points in our revised manuscript (P4 L18-22, and P15 L10-19).

**R1C 4:** On hitting the prior bound of residual uncertainty
Fig 1f clearly shows that the calibration tries to increase the residual variance and that high residual variances yields lower cost. In the current inversion, the residual variance is only bounded by the prior. This hints to deficiencies in the inversion. I sometimes experienced the same effect because a single equation of the cost (eq. 5) may in some cases not prefer the best fitting variance but the larger variance together with suboptimal parameters. Prescribing an upper bound is to my opinion not a good solution for this problem. Even fixing the residual variance would be a better option. My recommendation is to use several parameter blocks in a Metropolis within Gibbs sampling (Chib S Greenberg E (1995) Understanding the Metropolis-Hastings algorithm): One block to fit the model parameter conditional on the parameters of the residual statistical distribution and another block to fit the residual distribution parameters conditional on the current sample of model parameters.

**A1C 4:** We would like to clarify that SD in Fig 1 of the manuscript (both previous and revised manuscript) is not the residual variance. SD represents the effective soil depth and this is considered in this study as one of the parameters of BIOME-BGC simulator. In the figure cation, we have written that "Information about the BIOME-BGC parameters is given in Table 1". We have specifically mentioned in the figure caption that "SD is effective soil depth" in the revised manuscript to avoid any confusion.

Further Concerns of Referee #1

**R1C 5:** The cut of the posterior by the edge of the prior distribution of LFRT and FRC:LC (Fig 1) shows inconsistency in the combination of the model, the prior knowledge, and the observations. This hints to deficiencies of the calibration. It also makes it difficult to interpret the parameter estimates and process understanding. This needs more discussion. The introduction of bias parameters in model drivers or model predictions could help to resolve the inconsistencies and, moreover, the bias parameters then can be interpreted.

**A1C 5:** Fig 1 shows the cut of the posterior by the edge of the prior distribution, which is called edge effect, of LFRT and FRC:LC. We agree that this clearly indicates a significant effect on our posterior parameter space of LFRT and FRC:LC by our particular choices of parameter space to be included in the prior distributions of LFRT and FRC:LC. It could be argued that the prior uniform distributions of LFRT and FRC:LC could be made wider in order to eliminate the edge effect. However, we did not do this for the following reasons:

A.  We carried out extensive literature review in a previous study (Raj et al., 2014) to compile the information on FRC:LC, LFRT, and other BIOME-BGC parameters. This information led to the characterization of uncertainty in the parameters and helped to define the prior distributions. For the present study, we used this information on prior distributions. We had no further scope to make the prior distributions wider at the study site.

B.  In the present study, we have used the upper limit of FRC:LC at 2.15. In our previous study (Raj et al., 2014) we found that higher values of FRC:LC led to disappearance of the forest (LAI=0) due to negative cumulative NEE, and hence no production at the study site. Therefore, we had no other choice of the upper limit of FRC:LC other than 2.16.

Further, we don't fully agree that the edge effect indicates deficiencies of the calibration. This can also be thought in another way that even if there is the edge effect, a drop in RMSE and enhancement in NSE coefficient (Table 3) before and after calibration indicated the efficiency of the calibration.  We have mentioned above points in the revised manuscript (P10 L21-29, and P14 L5-7).

**R1C 6:**  How were the initial states of the model prescribed?

**A1C 6:** Initial states of the model were prescribed with very low value ($\approx 0$).  Spin up simulation of BIOME-BGC was performed first to achieve steady state condition of soil carbon and nitrogen pools under given climate and site condition. Normal simulation was then started with these steady state condition using daily meteorological data of 2009. We have added these points in the revised manuscript (P4 L7-10).

**R1C 7:**  Do you have correlations in the posterior parameter distribution, and how to you interpret them?

**A1C 7:**  In the revised manuscript, we have added a brief explanation and a plot showing the correlations in the posterior parameter distribution obtained in Experiment 1 (Fig 3 and P10 L30-32, and P11 L1-8). We found a strong positive correlation between the posterior distributions of C:Nleaf (carbon and nitrogen ratio) and FLNR (Fraction of leaf N in Rubisco) with r=0.95 (r is correlation coefficient). This strong positive correlation is in-line with the formulation of FLNR that shows direct proportionality with C:Nleaf (see Appendix A in RajEtal2014, for details). The parameters C:Nleaf and FLNR showed similar negative, but weak (> -0.5), correlation with Wint (Canopy water interception coefficient) ($r \approx -0.4$). This can be explained by the fact that the simulated GPP is expected to vary inversely with  Wint via soil water potential and stomatal regulation and directly with FLNR and C:Nleaf (see Section 5.1 in the previous or revised manuscript, for details of BIOME-BGC internal routines). The parameter SD (effective soil depth) had similar positive, but weak (< 0.5), correlation with FLNR and C:Nleaf ($r \approx 0.4$). This can be explained by the fact that the simulated GPP is expected to vary directly with SD (via soil water potential and stomatal regulation), and FLNR and C:Nleaf .Two parameters of any other pair combinations did not show any notable correlation.

**R1C 8:** Please discuss your finding in the context of other studies that already performed a Bayesian calibration of BGC-models against Flux data. E.g. there is big body of studies using the DALEC model also looking at multiple constraints, model error, and different sources uncertainties.

**A1C 8**: We couldn't find papers on a Bayesian calibration of BIOME-BGC against flux data that compare with our results directly. We found other papers on calibration of BIOME-BGC, and one

study (Hidy et al., 2012) on Bayesian calibration of BIOME-BGC.  Because Hidy et al. (2012) focussed on an herbaceous ecosystem, we could not compare directly the outcome quantitatively. .

Technical comments of Referee #1

**R1C 9:** Fig 1: Shows a very slow mixing. One chain needs more than 1000 steps to become uncorrelated with its previous state. Before computing the Gelman-Rubin criterion you should thin the chains by a factor so that autocorrelation or spectral density of the chain gets small.

**A1C 9:** We agree with the referee that thinning would reduce the spectral density of the chains. We decided not to do it again for the revised manuscript because we expect that thinning will not change the posterior estimations achieved in the study.

**R1C 10:** Fig 1: shortly explain phi and SD in the figure caption, e.g. "parameters describing variance and correlations of the distribution of model-data residuals (eq. 5)"

**A1C 10**: Please refer to our comment **A1C 4** on the clarification of SD. We have explained phi in the figure caption in the revised manuscript.

**R1C 11:** Fig 1: Maybe mention, that only the end of the chains after the burnin are shown**.**

**A1C 11:** Thank you very much for this suggestion. We have mentioned this in the figure caption in the revised manuscript.

**R1C 12:** Fig 3,. . .: Legends are missing. Please, use a different line type so that model and observations can be distinguished without color. Readers would benefit if you indicate months at the time axis instead of or in addition to Julian day.

**A1C 12:**  We have modified the figure in the revised manuscript according to the referee's suggestion.

**R1C 13:**  Fig 6 and associated discussion: For a model with state variables or pools this result is trivial. I suggest omitting or explicitly elaborating on the magnitude of the impacts of state versus drivers on the model output and with witch conditions the one or the other becomes important.

**A1C 13:**  We have omitted figure 6 in the revised manuscript.

**R1C 14:**  P9L25ff: More discussion needed on hitting the upper prior boundaries and its consequences.

**A1C 14:** Please refer to our comment **A1C 5.**

**R1C 15:** P10L18: typo percentile

**A1C 15:**  We have corrected this in the revised manuscript.

**R1C 16:** I cannot agree to the discussion because of the method that actually did not alter parameters across seasons during a single simulation run.

**A1C 16:** Please refer to our comment **A1C 2.**

**Response to Referee #2 for "Bayesian integration of flux tower data into process-based simulator for quantifying uncertainty in simulated output" by R. Raj et al.**

We thank for the constructive and helpful comments for our manuscript. We have carefully considered all comments and these have incorporated in our revised manuscript accordingly. We have inserted our response to each comment. We use "R2C" for referee #2's comment and "A2C" for author's response to referee #2.

**R2C 1:** The main problem of this manuscript is the use of time-varying parameters. The authors themselves recognise this as a problem (see page 13, lines 11/12). If I understand their use of time-varying parameters correctly ('engineering' a times series of GPP based on independent monthly sub-time series) it actually violates Bayes theorem, mass conservation and model dynamics. Of course one can do such an experiment to better understand the model dynamics and identify missing or misrepresented processes, but the authors are not taking this step and analysing the consequences of their results with the time-varying parameters in terms of model structure and formulation.

**A2C 1:** We understand the concern of referee about the use time varying parameters.  We found that the response of simulated GPP to weather conditions is rather similar among months: The simulated GPP was mainly driven by the meteorology, and much less by seasonal phenology.  We then hypothesized that some important state variables (such as LAI and carboxylation capacity) may not have a pronounced seasonal cycle in the model. In the second experiment, we calibrated Biome-BGC to the GPP of each month separately, as if the data for the other months did not exist. In that way neither mass conservation nor the Bayes theorem is violated. If some of the parameters have different optimum values when calibrated to different months of data, then this is an indication that the relation between these parameters and important state variables that (should) change during the season, may require improvement. The problem only arises when we combine the results in a time series. We then merge different simulations outputs into one. The objective was indeed to better understand the dynamics (or lack of dynamics) of the model. We decided to avoid the term 'time varying parameters' in the revision. We have mentioned above points in the revised manuscript (P14 L15-24, and P14 L25-35, and P15 L1-9).

**R2C 2:** Another concern is the use of GPP derived from eddy-covariance flux measurements as the observations in the calibration process. Eddy-covariance towers measure the net exchange flux, essentially NEE, and GPP is the derived from this net flux by employing a model. So essentially, the authors calibrate the BIOME-BGC parameters against another model, in this case the NRH model which makes its own assumptions about the dependency of GPP on environmental conditions.

**A2C 2:**  We would like to mention following points on using partitioned GPP instead of NEE data in this study:

A.  Indeed in our approach, the output of the process-based simulator was validated against the output of another model, notably the flux partitioning model. Although this approach has been used in other studies to validate the output of process-based simulator as well (Collalti et al., 2016, Liu et al., 2014, Yuan et al., 2014, Zhou et al., 2016), it could lead to error propagation. We clarify that the flux partitioning model (NRH model, Raj et al., (2016)) was tuned to the Eddy Covariance data in blocks of 10 days. Because the NHR and the

relationship of respiration with temperature and moisture were tuned for these short blocks separately, we expect that the GPP still reflects realistic responses to environmental drivers, and does not depend much on model assumptions.

B. Calibration of BIOME-BGC using NEE data is more challenging as NEE is the difference between fluxes caused by two processes (assimilation and respiration) We argue that calibration of such a complex model to NEE *instead* of GPP may not be a good idea, but calibration to NEE or respiration in addition to GPP is possible. However, we limited this study to the primary productivity, because this was our primary interest. A future study should be done to include both GPP data and ecosystem respiration data (can also be achieved by partitioning of NEE data) in a Bayesian calibration of BIOME-BGC. This may ensure the accuracy of all related carbon budget terms (GPP, NEE, and respirations).

We have discussed all above points in the revised manuscript (P4 L18-22, and P15 L10-19).

**R2C 3:** The whole Section 4.4 is not needed and does not provide any new insights, it is obvious that a dynamical model with state variables such as BIOME-BGC then also depends on its state variables.

**A2C 3:** We have removed section 4.4 in the revised manuscript.

**R2C 4:** So essentially the remaining part of the manuscript concerns experiment 1 and becomes rather light-weighted as a thorough analysis of the results from experiment 1 is missing. For example, how does the posterior error covariance matrix look like and what consequences does this have on the parameters (identifiability) and model? How does the posterior uncertainty compare to prior uncertainty? What is the impact of the observations on other simulated quantities (NEE, NPP), both in terms of their mean and uncertainty? How does the variability and the temporal autocorrelation compare to the prior?

**A2C 4:**
A. In the revised manuscript, we have added a brief explanation and a plot showing the correlations in the posterior parameter distribution obtained in Experiment 1 (Fig 3 and P10 L30-32, and P11 L1-8). We found a strong positive correlation between the posterior distributions of C:Nleaf (carbon and nitrogen ratio) and FLNR (Fraction of leaf N in Rubisco) with r=0.95 (r is correlation coefficient). This strong positive correlation is in-line with the formulation of FLNR that shows direct proportionality with C:Nleaf (see Appendix A in RajEtal2014, for details). The parameters C:Nleaf and FLNR showed similar negative, but weak (> -0.5), correlation with Wint (Canopy water interception coefficient) (r ≈ -0.4). This can be explained by the fact that the simulated GPP is expected to vary inversely with Wint via soil water potential and stomatal regulation and directly with FLNR and C:Nleaf (see Section 5.1 in the previous or revised manuscript, for details of BIOME-BGC internal routines). The parameter SD (effective soil depth) had similar positive, but weak (< 0.5), correlation with FLNR and C:Nleaf (r ≈ 0.4). This can be explained by the fact that the simulated GPP is expected to vary directly with SD (via soil water potential and stomatal regulation), and FLNR and C:Nleaf .Two parameters of any other pair combinations did not show any notable correlation.

B.  We have already compared posterior and prior uncertainty in the section 4.2 of the previous manuscript (Fig 2 and P9 L22-25).

C.  As far as the impact of the observations on other simulated quantities (NEE, NPP) is concerned, this is out of the scope of the present study. We mainly focused on the simulated GPP and already presented a lot of results.

D.  This study modelled the temporal correlation in the residuals during the calibration by adding the nuisance parameter φ in the likelihood function (see Section 3.3.2 in the previous manuscript). We had assumed uniform prior distribution of φ between -1 and +1. In the posterior, we obtained the range of φ from 0.56 to 0.93 with a mean at 0.75 (Fig 1g and P9 L15-18 in the previous manuscript). This showed the reduction in posterior uncertainty in φ compared to prior. We have now mentioned the choice of prior uncertainty in φ in the revised manuscript (P7 L19-20).

**R2C 5:** Also the terminology used in the manuscript is somewhat confusing. Sometimes the authors refer to simulated, sometimes to predicted GPP and sometimes to predicted flux tower GPP. In that context they also use the phrase 'posterior flux tower GPP', it is not clear to what the posterior refers?

A2C 5: We apologize for this confusion. We would like to clarify that the term "posterior" refers to the GPP obtained with posterior distribution of parameters. We agree that we had not used the terminology consistently. In the revised manuscript, we have made the terminology consistent as mentioned below:
   a.  "Flux tower GPP" - We have now used this single term throughout the manuscript to indicate GPP partitioned from flux tower observation of net ecosystem exchange.
   b.  "Posterior GPP" - We have now used this single term throughout the manuscript to indicate GPP simulated by BIOME-BGC at the posterior distribution of parameters.
   c.  "Prior GPP" - We have now used this single term throughout the manuscript to indicate GPP simulated by BIOME-BGC at the prior distribution of parameters.
   d.  "Simulated GPP" – Sometimes, we have used this term in the manuscript to give the general description of GPP simulated by BIOME-BGC irrespective of GPP simulated at prior or posterior distributions.
We have clarified this in the revised manuscript (P4 L17-18, P8 L12, P9 L26).

[revised manuscript text omitted]

---

## Editor Decision (ED1)

**Response to Authors**

Dear Rahul Raj and co-authors,

I thank you for your responses to my last comments on the article "Bayesian integration of flux tower data into process-based simulator for quantifying uncertainty in simulated output". You have answered, to a large extent, to the initial concerns I had about your modifications to answer the comments of the two reviewers. However before publishing this article I would like you to consider and take into account the following remarks (given that some of your modifications are not complete enough or/and not clear).

1) INTRODUCTION

For the justification of using only GPP data and not NEE, you added:
*"In principle, NEE data could be used alone to calibrate BIOME-BGC, where NEE is derived as the difference between the GPP and ecosystem respiration (Reco). A calibration of BIOME-BGC using NEE data only means that the difference between GPP and Reco equals the difference **between the two measured in the field** (Mitchell et al., 2011). Hence, the accuracy of simulated GPP can not be achieved using the NEE data alone."*
Such statement is slightly misleading as it seems to indicate that the GPP and RES are measured in the field; while you clearly explain later in the method that the GPP is only the result of a the NRH model. Please consider reformulating this added change to avoid any misleading interpretation.

At the end of the introduction, you state:
« The main novelty of this paper is the presentation of a Bayesian framework for BIOME-BGC parameters estimation. »
This statement is a bit restrictive and you should already mention that the paper also intend to provide some perspectives for other process-based models with respect to parameter calibration.

2) METHOD
Equation 6: typo with two signs "+ -"

3) SECTION 5.2 "BIOME-BGC calibration"

The sentence at the end of the first paragraph is not clear.
« The fact that the temporal correlation in the residuals is not only responsible for the temporal development of GPP indicated that the representation of dynamic processes within the simulator could be improved. »
Please try to be more explicit.

Third paragraph of section 5.2, you state:

« The resulting time series has discontinuities in state variables that can help to analyse the simulator behaviour in more detail. »

I don't understand why the discontinuities in state variables CAN HELP to analyze the simulator behavior; i.e. compared to a case where you would not have these discontinuities (with varying parameter along the simulation as initially suggested by the reviewers and not implemented). Please explain why or consider re-phrasing.

Added text in the third paragraph of section 5.2:

"This approach, however, can not be implemented in the original configuration of BIOME-BGC, ….. Such a modification in BIOME-BGC code is outside the scope of current study,.. »

The justification may sound weak, as modifying the code to include the possibility of varying the parameter values over time seems a "feasible task". You may consider adding that such implementation was not desired to illustrates that without code modification you can still investigate model structural errors (through varying parameter across the season) with your proposed approach 2.

End of third paragraph (added text):

You say : « If some parameters have different optimum values when calibrated against different months of data, then this indicates that the relation between these parameters and the state variables that (should) change during the season, may require improvement in the future. »

Please consider rewriting, as the sentence is not clear. Such sentence could also be more informative or points to the next paragraph that provides more insights on the potential of temporally varying parameters.

Note also that in the next paragraph, we miss an explanation on which parameter changes in experiment 2 are responsible for the reduction of the GPP in April (reduction to match the fluxnet derived data)?

Fith added paragraph :

You say : « We have, thus, provided as a general message that the temporal variation in the input parameters should receive further attention to the modelling communities focusing on simulating forest carbon cycle. »

Again this sentence is relatively vague! You could inform the reader about potential shortcoming of BIOME-BGC that may apply to other models; for instance those linked to LAI, VCMAX temporal evolution and the potentially missing processes that can be deduced from your experiment 2. This would reinforce the "messages or perspectives" for other similar models that the study brings. The objective is not to have varying parameters over time (then we can call them parameters) but to include all processes/equations to improve the temporal variations of state variables.

4) CONCLUSIONS (SECTION 6)

Point 2 about the experiment with/without the nuisance parameter "Phi". This part of the study is relatively novel and still poorly highlighted. In the conclusion you should try to expand a bit on the benefit for other modeling groups to include or not such term ; and thus to provide more general recommendations.

Point 3: the last sentence is relatively vague; it should summarize for process-based ecosystem modelers what you may learn in terms of model deficiencies with an experiment like your experiment-2

I apologize for the long delay in treating your responses.
Best regards,
Philippe

---

## Author Response (AR2)

Dr. Rahul Raj

Faculty of Geo-Information science and Earth Observation (ITC), University of Twente,

PO Box 217, 7500AE, Enschede, The Netherlands

Email: r.raj@utwente.nl

Date: 19 July 2017

Dear Philippe Peylin,

We are pleased to submit the second revised version of our manuscript titled "Bayesian integration of flux tower data into process-based simulator for quantifying uncertainty in simulated output". We are thankful for raising some more comments in our previous revised manuscripts. This helped us to present our research in a clearer way. In the revised manuscript, new text is highlighted in blue. Deleted text is indicated by strikethrough. We have attached our revision at the end of this letter. Our second revised version is the modified version of the revised manuscript that we submitted before. Our response (indicated by "AR" and bold text) to each comment (indicated by regular text) is given below.

Reviewer #1 (and #2): About the use of time varying parameters: Although your experiment 2 is well defined, the request from reviewer #1 of having a simulation with time varying parameters sounds justified. You mention "This approach, however, is not possible in BIOME-BGC because it is a dynamic vegetation simulator". Such answer is not obvious to understand. You could indeed increase the size of your "vector of parameters to be optimized" in order to have one parameter per month or proceed as proposed by reviewer #1 without significant model changes. This could indeed help to possibly identify what are the missing processes in the model controlling the temporal variation linked to the LFRT, FRC:LC, C:Nleaf, FLNR parameters at least. More generally the implication for model deficiencies that such experiment could highlight (either your current experiment 2 or a "true" time varying parameter experiment) is still not fully investigated in the new version.

**AR1: Thank you very much for your suggestion on the use of time varying parameters. We agree that our statement "This approach, however, is not possible in BIOME-BGC because it is a dynamic vegetation simulator" is not obvious to understand. For more clarification, we would like to mention the following:**

1. **Reviewer # 1 suggested that we could do either of the two procedures: (a) "A more feasible approach is to calibrate each month separately. For the next month calibration continues from a state of the previous month"; and (b) "For the currently used method, at minimum, the forward runs that produce the predictive posterior and the fit statistics need to change the parameters across months in each single forward run to discuss seasonally changing parameters". The procedure "a" could be implemented, only if it was possible to run BIOME-BGC separately for a single month in a given year. Then the next month calibration could actually be continued from the state of the previous month. This approach, however, cannot be implemented in the original configuration of BIOME-BGC. This is because a single forward run of BIOME-BGC simulates output for at least one year and the simulation can't be discontinued for a single month. The procedure "b" could be implemented, only if it was possible to change the parameters across the months during BIOME-BGC simulation for one year (note again, one year is the minimum period of simulation). This is, however, not possible in the original configuration of BIOME-BGC because only one set of parameter**

values is accepted. The implementation of both procedures "a" and "b" would require changing the BIOME-BGC code so that it could accept change in parameter across months and it could also be run for one month. Such modification in BIOME-BGC code is outside the scope of current study, which mainly demonstrated that the variation in the input parameters could be captured by means of Bayesian calibration.

In Experiment 2, we deliberately tempered with the BIOME-BGC simulator after we assumed that some important state variables (such as leaf area index (LAI) and carboxylation capacity (Vcmax)) may not have a pronounced seasonal cycle in the model. The reason behind this assumption is due to the overestimation of simulated GPP in the month of April (Fig 4 in the revised manuscript) with the constant calibrated parameters. We therefore calibrated the BIOME-BGC to the data of each month separately, as if we had no information on GPP for the other months. If some of the parameters have different optimum values when calibrated to different months of data, then this is an indication that the relation between these parameters and important state variables that (should) change during the season, may require improvement.

Even if we were not able to implement procedures "a" and "b" as suggested by reviewer #1, our Experiment 2 indicated important information about the consideration of variation of parameters in the future study by modifying the original configuration of BIOME-BGC.

2. For the next concern "model deficiencies that such experiment could highlight is still not fully investigated", we would like to emphasize two points. First, our manuscript already has a section 5.1 with Figure 7. This section helps the readers to establish the clear link between the six parameters (considered for calibration in this study) and the simulated GPP through the internal routines of BIOME-BGC. Second, in the section 5.2, we explained the possible reasons for the improvement of simulated GPP in Experiment 2 in the view of BIOME-BGC internal routines and two important state variables (LAI and Vcmax). A seasonality in the simulated GPP, which is directly linked to the seasonality in LAI and Vcmax, was more pronounced in Experiment 2 (with monthly variation in the posterior distributions of parameters) compared to Experiment 1 (without monthly variation in the posterior distributions of parameters). This was what we wanted to highlight as the deficiency of BIOME-BGC in simulating the seasonality of state variables without the variation in parameters during simulation.

We have clarified the above points in our revised manuscript (P2 L33-35, P3 L1-3, P3 L7-8, P14 L19-27, P15 L5-7).

About use of NEE data to calibrate the model: Your answer "We argue that calibration of such a complex model to NEE instead of GPP may not be a good idea but calibration to NEE or respiration in addition to GPP is possible" sounds not fully justify and partially answer the reviewer's request (given also that many studies have used NEE flux data to calibrate similar models)

AR2: We would like to give more clarification on this issue.

A major metrics of carbon cycle include GPP, ecosystem respiration ($R_{eco}$) and NEE. GPP is the first entry of atmospheric carbon into the ecosystem via the process of photosynthesis. NEE is derived as the difference between the GPP and $R_{eco}$. BIOME-BGC also simulates GPP and $R_{eco}$ first and then NEE is derived. This means that the errors in the simulated GPP and $R_{eco}$ propagate to NEE. A

calibration of BIOME-BGC against NEE data may ensure the accuracy of simulated NEE, however it does not guarantee the accuracy of simulated GPP and $R_{eco}$. The study of Tang et al. (2009) showed the strong improvement (by the drop of RMSE) in simulated GPP and $R_{eco}$ by the process based simulator TEM (Terrestrial Ecosystem Model) when both GPP and Reco data were used in the calibration compared to using NEE data alone. A calibration of PBS using NEE data only means that the difference between the two (GPP and $R_{eco}$) is equal the difference between the same two measured in the field (Mitchell et al., 2011). This indicates that the accuracy of simulated GPP and $R_{eco}$ can't be achieved using the NEE data alone in the calibration. The previous studies ( Fox et al., 2009; Kuppel et al., 2012) that used NEE data to calibrate process-based simulators DALEC (Data Assimilation Linked Ecosystem Carbon) and ORCHIDEE (ORganizing Carbon and Hydrology In Dynamic EcosystEms), therefore, were more successful in achieving the accuracy of this simulated difference compared to GPP and $R_{eco}$. Our study focused mainly on achieving the accuracy of simulated first entry of atmospheric carbon, i.e., GPP into the ecosystem and therefore we used partitioned GPP to calibrate BIOME-BGC simulator. This approach could, however, be extended to include $R_{eco}$ data together with NEE data in order to ensure the accuracy of all simulated metrics of carbon cycle. Then the parameters other than the six parameters considered in this study ,which may influence the simulated $R_{eco}$, need to be identified and should be included in the calibration.

We have clarified the above points in our revised manuscript (P1 L22-23, P2 L24-29, P15 L23-31 ).

Reviewer #1 ask your to discuss your finding in the context of other studies that already performed a Bayesian calibration of BGC-models against flux data. You have probably misunderstood such concern as indeed the reviewer does not refer to BIOME-BGC model but to BioGeoChemical models in general(BGC-models). He mention for example DALEC model but there are many other studies using FluxNet data based on model of similar complexity (JULES, ORCHIDEE, CABLE, LPJ, …). Please replace your study in the context of previous finding in order to bring specific conclusion that are relevant not only for the users of BIOME-BGC.

Reviewer #2: About his concern that a "thorough analysis of the results from Experiment 1 is missing". You made a substantial effort to analyze more in depth the results but the findings are still fully put in perspective with previous studies (i.e. to draw general messages for modelers in terms of Data Assimilation or Forest C-Cycle modeling).

AR3: As far as we understood, your both concerns (given by two paragraphs just above) are related to the same issue, i.e., indicating some general messages to the modellers in the revision. We would like to emphasize some points on this issue. For the process-based simulators, the dynamical processes within the simulators depend on the input parameters that are constant in time. The previous studies have also highlighted the improvement in the performance of simulator BEPS (Mo et al., 2008) and ORCHIDEE (Williams et al., 2009) with varying the input parameters over time. Those studies provided insight to the poorly understood dynamical processes related to photosynthetic capacity. In our study, we re-examined the variation in the input parameters of BIOME-BGC in a Bayesian framework related to photosynthetic capacity. We have, thus, provided as a general message that the temporal variation in the input parameters should receive further attention to the modelling communities focusing on simulating forest carbon cycle.

We have mentioned the above points in our revised manuscript (P15 L9-15, P16 L26 ).

Finally as a personal note: your change of the colors for figures 4-5-6 does not make enough distinction between the two "grey zones" (for the model and the data). Your initial colors choice was more clear.

**AR4: Thank you very much for this suggestions. In the revised manuscript, I have used initial colours choice.**

We look forward to hearing from you.

Yours sincerely

Rahul Raj
(on behalf of all authors)

**References:**

[revised manuscript text omitted]

---

## Author Response (AR3)

Dr. Rahul Raj

Faculty of Geo-Information science and Earth Observation (ITC), University of Twente,

PO Box 217, 7500AE, Enschede, The Netherlands

Email: r.raj@utwente.nl

Date: 07 November 2017

Dear Philippe Peylin,

We are pleased to submit the third revised version of our manuscript titled "Bayesian integration of flux tower data into process-based simulator for quantifying uncertainty in simulated output". We are thankful for raising some more comments in our previous revised manuscripts. This helped us to present our research in a clearer way. In the revised manuscript, new text is highlighted in blue. Deleted text is indicated by strikethrough. We have attached our revision at the end of this letter. Our third revised version is the modified version of the revised manuscript that we submitted before. Our response (indicated by "AR" and bold text) to each comment (indicated by regular text) is given below.

1) INTRODUCTION

For the justification of using only GPP data and not NEE, you added:

*"In principle, NEE data could be used alone to calibrate BIOME-BGC, where NEE is derived as the difference between the GPP and ecosystem respiration (Reco). A calibration of BIOMEBGC using NEE data only means that the difference between GPP and Reco equals the difference between the two measured in the field (Mitchell et al., 2011). Hence, the accuracy of simulated GPP can not be achieved using the NEE data alone."*

Such statement is slightly misleading as it seems to indicate that the GPP and RES are measured in the field; while you clearly explain later in the method that the GPP is only the result of a the NRH model. Please consider reformulating this added change to avoid any misleading interpretation.

**AR1: For clarity, we have rephrased the sentences in our revised manuscript (P2 L22-25) as follows:**

***"In principle, NEE data could be used alone to calibrate BIOME-BGC, where NEE is derived as the difference between the GPP and ecosystem respiration (Reco). Hence, a calibration of BIOME-BGC using NEE data only ensures the accuracy of difference between GPP and Reco (Mitchell et al., 2011). The accuracy of simulated GPP can not be achieved using the NEE data alone."***

At the end of the introduction, you state:

« The main novelty of this paper is the presentation of a Bayesian framework for BIOME-BGC parameters estimation. »

This statement is a bit restrictive and you should already mention that the paper also intend to provide some perspectives for other process-based models with respect to parameter calibration.

**AR2: Thank you for this suggestion. We have now added the following sentence in our revised manuscript (P3 L5-7):**

***"Additionally, investigation of temporal variation in BIOME-BGC input parameters would also reinforce to reconsider the assumption of constant parameters of other process-based simulators for photosynthesis."***

2) METHOD

Equation 6: typo with two signs "+ -"

**AR3: Thanks for pointing out this error. We have corrected this in our revision.**

3) SECTION 5.2 "BIOME-BGC calibration"
The sentence at the end of the first paragraph is not clear.
« The fact that the temporal correlation in the residuals is not only responsible for the temporal development of GPP indicated that the representation of dynamic processes within the simulator could be improved. »
Please try to be more explicit.

**AR4: In the revision, we have rephrased (P13 L30-34, P14 L1-3) a part of first paragraph in section 5.2 to make the statement more explicit. Below I provide the whole paragraph with rephrased sentences:**

*"The posterior GPP closely followed the flux tower GPP even for those months (September and October) which were not included in the calibration (Fig. 5), although this was not perfect as shown by the fact that $\phi \neq 0$. If the simulator would properly capture the temporal development of GPP we would expect that $\phi = 0$, even after allowing for some uncertainty in the prediction. We deliberately assumed $\phi = 0$ in the likelihood function (Eq. 6) to check if this assumption has any effect on the posterior GPP. We, however, found that both choices $\phi \neq 0$ and $\phi = 0$ led to similar posterior GPP (section 5.3). This comparison indicated that an improvement in temporal development of GPP after calibration might not be achieved, at least for BIOME-BGC simulator, with either the assumption of presence or absence of temporal correlation in the residuals. The representation of dynamic processes within the simulator responsible for GPP should be, therefore, given more attention in order to improve the temporal development of GPP. This is what we showed in Experiment 2."*

« The resulting time series has discontinuities in state variables that can help to analyse the simulator behaviour in more detail. »
I don't understand why the discontinuities in state variables CAN HELP to analyze the simulator behavior; i.e. compared to a case where you would not have these discontinuities (with varying parameter along the simulation as initially suggested by the reviewers and not implemented). Please explain why or consider re-phrasing.

**AR5: For clarity, we have rephrased the sentences in our revised manuscript (P14 L17-20) as follows:**

*"The resulting time series has discontinuities in state variables and the update of simulator memory between the months is ignored. This time series can, however, help to analyse the simulator behaviour for the temporal variation in the input parameters."*

Added text in the third paragraph of section 5.2:
"This approach, however, can not be implemented in the original configuration of BIOMEBGC, ….. Such a modification in BIOME-BGC code is outside the scope of current study,.. »
The justification may sound weak, as modifying the code to include the possibility of varying the parameter values over time seems a "feasible task". You may consider adding that such implementation was not desired to illustrates that without code modification you can still investigate model structural errors (through varying parameter across the season) with your proposed approach 2.

**AR6: Thanks for the suggestion. We have added this in the revised manuscript (P14 L26-27) as follows:**

*"Such a modification was, however, not desired because model deficiency of BIOME-BGC could still be investigated through the temporal variation in the input parameters across the season using the approach proposed in Experiment 2."*

End of third paragraph (added text):
You say : « If some parameters have different optimum values when calibrated against different months of data, then this indicates that the relation between these parameters and the state variables that (should) change during the season, may require improvement in the future. »
Please consider rewriting, as the sentence is not clear. Such sentence could also be more informative or points to the next paragraph that provides more insights on the potential of temporally varying parameters.
Note also that in the next paragraph, we miss an explanation on which parameter changes in experiment 2 are responsible for the reduction of the GPP in April (reduction to match the fluxnet derived data)?

**AR7: For clarity, we have rephrased the sentence in our revised manuscript (P14 L31-33) as follows:**

*"If the obtained variations in the input parameters improve the seasonality in simulated GPP, this indicates that the default linkage of the constant parameters with the state variables, that change during the season, in the simulator may require improvement in future study."*

**We have added following sentence in the revised manuscript (P15 L5-6) to explain which parameters are responsible for the reduction in GPP in April:**

*"During the start of growing season (April), higher Wint and lower SD maintained low GPP rates."*

Fith added paragraph :
You say : « We have, thus, provided as a general message that the temporal variation in the input parameters should receive further attention to the modelling communities focusing on simulating forest carbon cycle. »
Again this sentence is relatively vague! You could inform the reader about potential shortcoming of BIOME-BGC that may apply to other models; for instance those linked to LAI, VCMAX temporal evolution and the potentially missing processes that can be deduced from your experiment 2. This would reinforce the "messages or perspectives" for other similar models that the study brings. The objective is not to have varying parameters over time (then we can call them parameters) but to include all processes/equations to improve the temporal variations of state variables.

**AR8: Thank you very much for this suggestion. For clarity, we have added following arguments in the fifth paragraph (section 5.2) of the revised manuscript (P15 L20-26):**

*"In our study, we re-examined the temporal variation in the input parameters, related to photosynthetic capacity, of BIOME-BGC in a Bayesian framework. We observed that the temporal dynamics of the state variables (LAI and Vcmax) and the soil water mechanism within BIOME-BGC, and thus photosynthesis, are not sufficiently expressed by the constant input parameters. These state variables also control photosynthesis simulations in other process-based simulators, such as*

*SCOPE (van der Tol et al., 2009), and are governed by the constant input parameters that may not be adequate based on our findings. Our study, therefore, reinforce a message that the reconsideration of temporal dynamics of state variables within the simulator, possibly through the temporal variation in the parameters, should receive further attention to the modelling communities focusing on simulating forest carbon cycle."*

4) CONCLUSIONS (SECTION 6)
Point 2 about the experiment with/without the nuisance parameter "Phi". This part of the study is relatively novel and still poorly highlighted. In the conclusion you should try to expand a bit on the benefit for other modeling groups to include or not such term ; and thus to provide more general recommendations.

**AR9: As per your suggestion, we have revised point 2 of the conclusion in the revised manuscript (P16 L21-26) to provide more general recommendations. Below I provide the revised point 2:**

*"We modelled the temporal correlation in the residuals through the nuisance parameter, $\phi$, in the likelihood function. We concluded that BIOME-BGC did not properly simulate the temporal development of GPP, neither by assuming temporal correlation in the residuals ($\phi \neq 0$) nor by ignoring this ($\phi = 0$) and the dynamical processes within the BIOME-BGC became more prominent. Hence calibration gave greater insight into the BIOME-BGC. Other future studies on the calibration of similar process-based simulators may also ignore $\phi$, but they should consider carefully the dynamic processes within the simulators to achieve improved calibration results."*

Point 3: the last sentence is relatively vague; it should summarize for process-based ecosystem modelers what you may learn in terms of model deficiencies with an experiment like your experiment-2.

**AR10:  We have revised point 3 of the conclusion in the revised manuscript (P16 L27-31, P17 L1-3) to summarize our findings for other ecosystem modellers. Below I provide the revised point 3:**

*"We used the calibration results to gain further insights into the functioning (dynamic processes) of BIOME-BGC through analysis of the monthly variation in posterior parameter distributions. Our study revealed the model deficiency of BIOME-BGC for using constant parameters to simulate seasonality of state variables, thus the seasonality in daily GPP. The seasonality was captured more precisely by using monthly variation in the BIOME-BGC parameters. In future, such model deficiency should receive attention by the BIOME-BGC modelling communities. Nevertheless, our findings also suggest that the other modelling communities that use the similar process-based simulators may also consider to improve such model deficiency."*

We look forward to hearing from you.

Yours sincerely
Rahul Raj
(on behalf of all authors)

**References:**

[revised manuscript text omitted]